# C-IFS-CB05-BASCOE: Stratospheric Chemistry in the Integrated Forecasting System of ECMWF

Vincent Huijnen[1], Johannes Flemming[2], Simon Chabrillat[3], Quentin Errera[3], Yves Christophe[3], Anne-Marlene Blechschmidt[4], Andreas Richter[4], Henk Eskes[1]

[1] Royal Netherlands Meteorological Institute, De Bilt, The Netherlands
[2] European Centre for Medium-Range Weather Forecasts, Reading, UK
[3] Belgian Institute for Space Aeronomy (BIRA-IASB), Brussels, Belgium
[4] Institute of Environmental Physics, University of Bremen, Germany

*Correspondence to*: Vincent Huijnen (Huijnen@knmi.nl)

**Abstract.** We present a model description and benchmark evaluation of an extension of the tropospheric chemistry module in the Integrated Forecasting System (IFS) of the European Centre for Medium-Range Weather Forecasts (ECMWF) with stratospheric chemistry, referred to as C-IFS-CB05-BASCOE (for brevity here referred to as C-IFS-TS). The stratospheric chemistry originates from the one used in the Belgian Assimilation System for Chemical ObsErvations (BASCOE), and is here combined with the modified CB05 chemistry module for the troposphere as currently used operationally in the Copernicus Atmosphere Monitoring Service (CAMS). In our approach either the tropospheric or stratospheric chemistry module is applied depending on the altitude of each individual grid box with respect to the tropopause. An evaluation of a 2.5 year long C-IFS-TS simulation with respect to various satellite retrieval products and in-situ observations indicates good performance of the system in terms of stratospheric ozone, and a general improvement in terms of stratospheric composition compared to the C-IFS predecessor model version. Possible issues with transport processes in the stratosphere are identified. This marks a key step towards a chemistry module within IFS that encompasses both tropospheric and stratospheric composition, and could expand the CAMS analysis and forecast capabilities in the near future.

## 1 Introduction

Existing earth observation systems in combination with global circulation models (GCMs) help to provide a better understanding of the Earth's atmospheric composition and changes therein (Hollingsworth et al., 2008). For the troposphere, hemispheric transport and chemical conversion of atmospheric composition influences regional air quality (Pausata et al., 2012; Im et al., 2015, Marécal et al., 2015). Also, the amount of stratospheric ozone directly impacts the forecast capabilities of surface solar irradiance (Qu et al., 2014), stressing the relevance of good stratospheric ozone forecasts. Stratospheric ozone further affects the chemical composition in the troposphere because of stratosphere-troposphere transport of ozone (Stevenson et al., 2006, Gaudel et al., 2015), and its radiative properties influencing the tropospheric photolysis rates. Beyond such direct implications on the troposphere a comprehensive description of stratospheric composition allows a more complete understanding of processes taking place in the stratosphere, ranging from tracking the ozone hole (Lefever et al.,

2015) and understanding the concentrations of ozone depleting substances (Chipperfield et al., 2015), to the assessment of dynamical effects such as the Quasi-Biennial Oscillation (QBO, Baldwin et al., 2001), and from implications of sudden stratospheric warmings on circulation patterns (Manney et al., 2015) to general radiative feedbacks of ozone, water vapour and $CO_2$ on weather and climate (Solomon et al., 2010).

These aspects have long been studied in the framework of Chemistry Transport Models (CTMs) and, more recently, in GCMs, see, e.g., the SPARC Chemistry-Climate Model Validation Activity (CCMVal, 2010). In GCMs the role of stratospheric ozone chemistry on the tropospheric climate can explicitly be studied (e.g. Scaife et al., 2011). But also meteorological models can benefit from having a good representation of the stratospheric composition and its variability, considering the radiative effects and the resulting impact on stratospheric as well as tropospheric temperatures (Monge-Sanz

et al., 2013), which becomes relevant for tropospheric forecast skills on long-range to seasonal time scales (Maycock et al., 2011).

Within a series of MACC (Monitoring Atmospheric Composition and Climate) European research projects a global forecast and assimilation system has been built, which is the core for the global system of the Copernicus Atmosphere Monitoring Service, (CAMS, http://atmosphere.copernicus.eu ). In CAMS, forecasts of atmospheric composition are carried out

(Flemming et al., 2015, Morcrette et al., 2009, Engelen et al. 2009), which benefit from assimilation of satellite retrievals (Inness et al., 2015, Benedetti et al., 2009), to improve the initial conditions for composition fields in terms of reactive gases, aerosols and greenhouse gases. Here a tropospheric chemistry scheme has been embedded in ECMWF's Integrated Forecast System, referred to as Composition-IFS (C-IFS, Flemming et al., 2015). Even though the current operational version of C-IFS based on the Carbon Bond chemistry scheme (CB05) provides good model capability on tropospheric composition

(Eskes et al., 2015), the stratosphere is only realistically constrained in terms of ozone. This is because so far the model ozone is based on a linear scheme (Cariolle and Tyssèdre, 2007) which is suitable owing to the data-assimilation capabilities of C-IFS of both total column and profile satellite retrievals (Flemming et al., 2011; Inness et al., 2015; Lefever et al., 2015). Also it is recognized that the applicability of radiation feedbacks of trace gases, such as ozone and water vapour, as produced through $CH_4$ oxidation, are hampered by schemes that are based on linearizations (Cariolle and Morcrette, 2006; de Grandpré

et al., 2009), This is due to the intrinsic dependencies to climatologies which are used to construct such schemes and hence they may behave poorly in anomalous situations. Having full stratospheric chemistry available in the IFS therefore would not only allow to study a wider range of atmospheric composition processes, but also a more independent assessment of radiation feedbacks on temperature, hence providing the potential for improvements in stratospheric and tropospheric meteorology. These considerations drive the need for extension of C-IFS with a module for stratospheric chemistry. For this

we use the chemistry scheme from the Belgian Assimilation System for Chemical ObsErvations (BASCOE), Errera et al. (2008), which was developed to assimilate satellite observations of stratospheric composition.

BASCOE is based on a Chemistry Transport Model (CTM) of the stratosphere which is used to investigate stratospheric photochemistry (Theys et al., 2010; Muncaster et al., 2012). The assimilation system uses the 4D-VAR algorithm (Talagrand and Courtier, 1987) to produce reanalyses of stratospheric composition (Viscardy et al., 2010) which compare favourably

well with similar systems (Geer et al., 2006; Thornton et al., 2009) and facilitate detailed studies of transport processes in the stratosphere (Lahoz et al., 2011). The photochemistry module from the BASCOE-CTM was implemented into the Canadian assimilation system GEM, demonstrating the potential of a coupled chemical-dynamical assimilation system for stratospheric studies (de Grandpré et al., 2009; Robichaud et al., 2010). BASCOE has been used and evaluated within the

framework of MACC as an independent system for the provision of Near Real-Time analyses of stratospheric ozone and for the validation of the corresponding product by the main assimilation system (Lefever et al., 2015; Eskes et al., 2015).

The CB05 tropospheric scheme has been combined with the stratospheric scheme from BASCOE-CTM to form a single chemistry mechanism that encompasses tropospheric and stratospheric chemistry throughout the atmosphere, here referred to as C-IFS-Atmos. However, this approach appears computationally expensive, due to the extended chemical mechanism.

Therefore we have developed an approach for an optimized merging of the CB05 tropospheric chemistry scheme and the stratospheric chemistry scheme used in BASCOE within C-IFS. An assessment of the two chemistry schemes showed that there is only partial overlap in trace gases and reactions that are essential in both regimes. For instance, 15 out of the full list of 99 trace gases need to be treated in the chemical mechanisms for both troposphere and stratosphere. Also the modelling of the photolysis rates and heterogeneous reactions have been optimized for application in troposphere and stratosphere

separately. In this optimized approach we developed a flexible setup where -within a single framework- either the tropospheric or stratospheric chemistry modules are addressed, referred to as C-IFS-TS. In this approach the parameterizations for the chemistry, including the respective chemistry mechanisms as optimized for troposphere and stratosphere separately, are retained.

In this paper we describe two merging approaches and provide benchmark evaluations of the C-IFS-Atmos and C-IFS-TS

systems with focus on the stratospheric composition. The ancestor BASCOE-CTM is also included in the comparison through a forward model run (without chemical data assimilation), in order to provide insight in the differences caused by the treatment of transport between C-IFS and BASCOE. The paper is organized as follows: In Sect 2 the chemistry modules for the stratosphere are described and the merging with the tropospheric scheme is explained.. Section 3 provides details on the setup of the model runs, and the observational data used for the model evaluation. Section 4 provides a basic model

evaluation of the system. We finalize this manuscript with conclusions and an outlook for further work.

## 2. Atmospheric chemistry in C-IFS

For general aspects related to chemistry modeling in C-IFS the reader is referred to Flemming et al. (2015). The meteorological model in the current version of C-IFS is based on IFS cycle 41r1 (ECMWF, 2015). The advection is simulated with a three-dimensional semi-Lagrangian advection scheme, which applies a quasi-monotonic cubic interpolation

of the departure values.

In the following two subsections we describe the C-IFS modules for the stratospheric (BASCOE-based) and tropospheric (CB05-based) chemistry parameterizations, continued by a section describing the merging procedure of these two modules to

form the C-IFS-TS system. The full list of trace gases is given in Table A1 in the Appendix, including the domains where they are actively treated within the chemistry.

## 2.1 Stratospheric chemistry

From the BASCOE system (Errera et al., 2008) the chemical scheme and the parameterization for Polar Stratospheric Clouds
(PSC) has been implemented in C-IFS. The BASCOE chemical scheme used here is labelled "sb14a". It includes 58 species interacting through 142 gas-phase, 9 heterogeneous and 52 photolytic reactions. This chemical scheme merges the reaction lists developed by Errera and Fonteyn (2001) to produce short-term analyses, with the list included in the SOCRATES 2-D model for long-term studies of the middle atmosphere (Brasseur et al., 2000; Chabrillat and Fonteyn, 2003). The resulting list of species (see Table A1) includes all the ozone-depleting substances and greenhouse gases necessary for multi-decadal
simulations of the couplings between dynamics and chemistry in the stratosphere, as well as the reservoir and short-lived species necessary for a complete description of stratospheric ozone photochemistry.

Gas-phase and heterogeneous reaction rates are taken from JPL evaluation 17 (Sander et al., 2011) and JPL evaluation 13 (Sander et al., 2000), respectively. Lookup tables of photolysis rates were computed offline by the TUV package (Madronich and Flocke, 1999) as a function of log-pressure altitude, ozone overhead column and solar zenith angle. The photolysis
tables used in chemical scheme sb14a are based on absorption cross-sections from JPL evaluation 15 (Sander et al., 2006). The kinetic rates for heterogeneous chemistry are determined by the parameterization of Fonteyn and Larsen (1996), using classical expressions for the uptake coefficients on sulfate aerosols (Hanson and Ravishankara, 1994) and on Polar Stratospheric Clouds (PSCs) (Sander et al., 2000).

The surface area density of stratospheric aerosols uses an aerosol number density climatology based on SAGE-II
observations (Hitchman et al., 1994). Ice PSCs are presumed to exist at any grid point in the winter/spring polar regions where water vapour partial pressure exceeds the vapour pressure of water ice (Murphy and Koop, 2005).

Nitric Acid Tri-hydrate (NAT) PSCs are assumed when the nitric acid ($HNO_3$) partial pressure exceeds the vapour pressure of condensed $HNO_3$ at the surface of NAT PSC particles (Hanson and Mauersberger, 1988). The surface area density is set to $2 \times 10^{-6}$ $cm^2/cm^3$ for ice PSCs and $2 \times 10^{-7}$ $cm^2/cm^3$ for NAT PSCs. The sedimentation of PSC particles causes
denitrification and dehydration. This process is approximated by an exponential decay of $HNO_3$ with a characteristic time-scale of 20 days for gridpoints where NAT particles are supposed to exist, and an exponential decay of $HNO_3$ and $H_2O$ with a characteristic time-scale of 9 days for gridpoints where ice particles are supposed to exist.

Mass mixing ratios for $N_2O$, $CO_2$ and a selection of anthropogenic and organic halogenic trace gases are constrained at the surface by a global mean constant value, Table 1. Assuming that trace gases are well mixed in the troposphere, this
essentially serves as lower boundary conditions for the stratospheric chemistry.

## 2.2 Tropospheric chemistry

The tropospheric chemistry in the C-IFS is based on the CB05 mechanism (Yarwood et al., 2005). It adopts a lumping approach for organic species by defining a separate tracer species for specific types of functional groups. The scheme has been modified and extended to include an explicit treatment of C1 to C3 species as described in Williams et al., (2013), and SO$_2$, di-methyl sulphide (DMS), methyl sulphonic acid (MSA) and ammonia (NH3) (Huijnen et al., 2010). A coupling to the MACC aerosol model is available (Huijnen et al., 2014), but not switched on for this study. The reaction rates follow the recommendations given in either Sander et al. (2011) or Atkinson et al. (2006). The modified band approach (MBA), which is adopted for the computation of photolysis rates (Williams et al., 2012), uses 7 absorption bands across the spectral range 202 − 695 nm. At instances of large solar zenith angles (71-85°) a different set of band intervals is used. In the MBA the radiative transfer calculation using the absorption and scattering components introduced by gases, aerosols and clouds is computed on-line for each of the predefined band intervals. The complete chemical mechanism as applied for the troposphere is extensively documented in Flemming et al. (2015). There a specification of the emissions and deposition of tropospheric reactive trace gases is provided as well.

## 2.3 Merging procedures for the tropospheric and stratospheric chemistry

Here we investigate two options to merge tropospheric and stratospheric chemistry, as also summarized in Table 2. The chemistry mechanism for C-IFS-Atmos is composed by simply combining the reaction mechanisms for troposphere and stratosphere into one large mechanism, removing reactions that are duplicated. In contrast to this model version here we propose an approach for a more efficient merging of the chemistry modules for the troposphere and stratosphere to form the C-IFS-TS system. Key chemical cycles differ between troposphere and stratosphere, hence allowing different chemical mechanisms. For example, the oxidation of non-methane hydrocarbons (NMHC's) is essentially taking place in the troposphere and represents an important driver for tropospheric O$_3$ production. The chemical evolution of PAN and organic nitrate can be neglected in the stratosphere. On the other hand, N$_2$O and CFC's are essentially chemically inactive in the troposphere and will only be photolysed by UV radiation in the stratosphere. Therefore, the chemical reactions involving these gases do not need to be accounted for in the troposphere. . Also the parameterization of the photolysis rates leads to different requirements for the troposphere and stratosphere, as will be discussed in the next subsection. Finally the numerical solver of the chemical mechanism contributes substantially to the total costs of the model run in terms of run-time, depending on the size of the reaction mechanism. These elements have motivated us to divide the chemistry in the C-IFS-TS system into a tropospheric and stratospheric part. Note that there is only one set of transported atmospheric trace gases and only the position of the grid box above or below the tropopause determines if the tropospheric or stratospheric chemistry is applied.

The tropopause can be defined based on a different criteria. A common approach is to use dynamical criterion such as the isentropic potential vorticity (e.g., Thuburn and Craig, 1997) but this fails in regions of small absolute vorticity, notably in

the tropics. A definition based on the lapse rate (WMO, 1957) is an alternative, but may not be well defined in the presence of multiple stable layers. We therefore choose to base our criterion on the chemical composition of the atmosphere considering that the tropopause is associated with sharp gradients in trace gases (e.g., Gaudel et al., 2015). This has the advantage that parcels with tropospheric/stratospheric composition can be traced dynamically, and the most appropriate

chemistry scheme can be adopted to it. In our simulation we use a chemical definition of the tropopause level, where tropospheric grid cells are defined at $O_3<200$ ppb and $CO>40$ ppb, for $P > 40$ hPa. With this definition the associated tropopause pressure ranges in practice between approx. 270 and 80 hPa for sub-tropics and tropics, respectively.

For both troposphere (CB05) and stratosphere (BASCOE) the numerical solver is generated using the Kinetic Pre-Processor (KPP, Sandu and Sander, 2006) software. Specifically we adopt the standard four-stages, third order Rosenbrock solver

(Rodas3). This is different from the Eulerian backward implicit solver as used in Flemming et al. (2015), and is motivated by the improved coding flexibility and accuracy.

Most of the gas phase reactions that take place both in the troposphere and stratosphere, such as $NO_x$ and $HO_x$ reactions, are simulated in identical ways in both chemistry schemes. It is worth mentioning that the constituents $O^1D$ and $O^3P$, produced from $O_3$ and $O_2$ photolysis, are not explicitly computed in the troposphere, as $O^1D$ and $O^3P$ are assumed to react with $O_2$, $O_3$

and $N_2$ only. This is different for the stratosphere, where $O^1D$ and $O^3P$ are involved in many reactions. For trace gases whose chemistry is currently neglected in the stratosphere (the NMHC's, PAN, Organic nitrate, $SO_2$) we adopt a 10-day decay rate to prevent their spurious accumulation in the stratosphere. Hence these losses are currently not accounted for in the stratospheric chemical mechanisms and do not contribute either to the load of stratospheric aerosols. Note that tropospheric halogen chemistry, which contributes to near-surface ozone depletion in spring-time polar region and to

changes in oxidative capacity in the tropical marine boundary layer (von Glasow and Crutzen, 2007) is currently neglected, even though related trace gases are available. By inspection of individual constituents fields we have ensured that the merging strategy does not result in spurious jumps at the interface between troposphere and stratosphere, see also Supplementary Figures S2-S5. When the system is run with stratospheric chemistry only (C-IFS-S), all chemistry and emissions are switched off at altitudes below 400 hPa and constrained by surface boundary conditions.

The four options to run this type of C-IFS experiments and the computational costs are given in Table 2. As compared to the C-IFS-T experiments, the costs of running an experiment including full stratospheric chemistry with the C-IFS-TS system have increased by ~50%. Most of this increase is caused by the computation of the chemistry and not the tracer transport due to the efficiency of the semi-Lagrangian advection scheme for multiple tracers. The C-IFS-Atmos setup where tropospheric and stratospheric chemistry were merged into a single reaction mechanism, led to an increase in costs by ~50% compared to

C-IFS-TS, indicating the benefit of having separate solver codes for tropospheric and stratospheric chemistry. The C-IFS-TS implementation allows for an easy switch between system setups compared to the C-IFS-Atmos implementation. For completeness also specifications of the BASCOE-CTM are provided in Table 2, which is identical in terms of stratospheric chemistry parameterization compared to C-IFS-TS and C-IFS-S. Clearly the essential difference compared to the C-IFS

setup refers to the fact that BASCOE is used here as a CTM, while C-IFS is a GCM. Most notably this implies a different advection treatment and a different horizontal grid (see section 3).

### 2.3.1 Merging photolysis rates

For parameterization of the photolysis rates the Modified Band Approach (MBA, Williams et al., 2012) and the lookup table approach (Errera and Fonteyn, 2001) are retained, see Table 3, as these have been optimized in the past for applications in the troposphere and stratosphere, respectively. While for tropospheric conditions scattering and absorption properties of clouds and aerosol strongly impact the magnitude of photolysis rates and hence the local chemical composition, this is of less relevance in the stratosphere. Wavelengths shorter than 202 nm, on the other hand, are largely blocked by stratospheric ozone and oxygen and do not contribute to radiation in the troposphere (Williams et al., 2012). At higher altitudes these short wavelengths contribute to the Chapman cycle and to the break down of $CH_4$, CFC's and $N_2O$ either directly or through oxidation by $O^1D$. Also the presence of sunlight at solar zenith angles (SZA) larger than 90° at high altitudes needs to be accounted for in the stratosphere due to the Earth's curvature. This plays a role in the timing of springtime ozone depletion in the polar lower stratosphere, but may be neglected in the troposphere.

Table 4 lists the photolysis rates that are active both in the troposphere and stratosphere. Photolysis rates for reactions occurring both in the troposphere and stratosphere are merged at the interface, in order to ensure a smooth transition between the two schemes. This is done by an interpolation at four model levels around the interface level between both parameterizations, for SZA<85°. For larger SZA the original value for the photolysis rate is retained in case of stratospheric chemistry, while it is switched off for the troposphere.

Note that even though the reaction rates have been merged, the products from the same photolytic reactions are sometimes different as a consequence of the different reaction mechanisms between the troposphere and stratosphere.

An example of the merging strategy is given in Fig. 1. It shows that at the interface for J $O_3$ and J $NO_2$ on average a small increase of the merged photolysis rate is seen towards lower altitudes, with the switch to MBA in the troposphere, which is a consequence of the combination of differences in the parameterizations. Even though such jumps are undesirable, no visible impact on local chemical composition was found, for any of the trace gases involved in both tropospheric and stratospheric chemistry, see also Figures S1-S3 in the Supplementary Material. This can be explained by the sufficiently small difference in the photolysis rates at the merging altitude of the photolysis and chemistry schemes, combined with the sufficiently long lifetime of the affected trace gases.

### 2.3.2 Tracer transport settings

Tracer transport is treated identically for all individual chemical trace gases. Since the semi-Lagrangian advection does not formally conserve mass (Flemming and Huijnen, 2011; de Grandpré et al., 2016) a global mass fixer is applied (Diamantakis and Flemming, 2014) to all but a few constituents, including NO, $NO_2$ and $H_2O$. Rather than conserving mass during the advection step of the individual components NO and $NO_2$, this is enforced to a stratospheric $NO_x$ tracer defined as the sum of

NO and $NO_2$. While a chemical $H_2O$ trace gas is defined in the full atmosphere, in the troposphere $H_2O$ mass mixing ratios are constrained by the humidity (q) simulated in the meteorological model in IFS and provide a boundary condition for water vapour in the stratosphere. Stratospheric $H_2O$ (i.e. above the tropopause level) is governed by chemical production and loss. The global advection errors in $H_2O$ that essentially originate from the tropospheric part because by far most $H_2O$ mass is

located in the troposphere and the spatial gradients are much more pronounced. This should not affect the stratospheric $H_2O$ mass budget, herefore the global mass fixer for the stratospheric $H_2O$ tracer has been switched off. This prevents spurious trends in stratospheric $H_2O$ columns over the years (not shown), indicating that $H_2O$ mass conservation is well ensured in the stratosphere.

## 3. Model setup and observations used

We have executed runs with C-IFS-TS and C-IFS-Atmos for the period April 2008 until December 2010. Stratospheric ozone in C-IFS-TS is further compared to that of the C-IFS-T system (Flemming et al., 2015). This uses the ECMWF standard linear ozone scheme (version 2a, Cariolle and Teyssèdre, 2007) in the stratosphere, while stratospheric $HNO_3$ is constrained through a climatological ratio of $HNO_3/O_3$ at 10 hPa (Flemming et al., 2015).

We have initialized all C-IFS runs on 1 April 2008 using assimilated concentration fields from the BASCOE system in the

stratosphere for this date. The horizontal resolution of these runs is T255 (i.e. approx. 0.7° lon / lat) with 60 levels in the vertical. Meteorology in the C-IFS runs is relaxed towards ERA-Interim.

Intercomparison of the runs C-IFS-TS and C-IFS-Atmos aims to provide a justification of our approach to split the chemistry into two regions, while intercomparison of C-IFS-TS with C-IFS-T can be used to identify the changes to stratospheric composition modelling between full stratospheric chemistry and the baseline approach with the linear ozone scheme.

The performance of the C-IFS runs has further been compared against the BASCOE-CTM (without chemical data assimilation), using the same chemical mechanism and parameterizations for photolysis and heterogeneous chemistry as implemented in the C-IFS-TS. This serves as a model reference for the C-IFS implementation of stratospheric chemistry. While C-IFS evaluates tracer transport on a reduced Gaussian grid, the BASCOE-CTM uses a regular latitude-longitude grid. It is run here with a resolution of 1.125° lon / lat similar to the resolution chosen for C-IFS used, and on the same

vertical grid of 60 levels. The BASCOE-CTM is driven by temperature, pressure and wind fields simulated by the C-IFS runs. However, while BASCOE adopts a flux-form advection scheme (Lin and Rood, 1996) the IFS uses the Semi-Lagrangian scheme for advection, accounts for vertical diffusion and includes a parameterization for convection (ECMWF, 2015). Using essentially the same dynamical fields together with an identical implementation of the chemistry code should allow to identify differences due to the different transport schemes between C-IFS and the BASCOE-CTM. Common

chemical biases between both systems also point at issues in the chemical parameterizations such as reaction mechanism, photolysis, heterogeneous chemistry and sedimentation.

## 3.1 Observational data used for validation

We evaluate the C-IFS runs in terms of stratospheric $O_3$, $NO_2$, $N_2O$, $CH_4$, $H_2O$ and HCl, and for this purpose use a range of observation-based products.

Model results are compared with retrievals (version 3) of $O_3$, (Froidevaux et al., 2008a), ClO (Santee et al., 2008), $H_2O$ (Read et al., 2007) and HCl (Froidevaux et al., 2008b) from the Microwave Limb Sounder (MLS) onboard the satellite Aura and with retrievals (version 6) of $O_3$ (Ceccherini et al., 2008), $HNO_3$ (Wang et al., 2007) and $NO_2$ (Wetzel et al., 2007) from limb emission spectra recorded by the Michelson Interferometer for Passive Atmospheric Sounding (MIPAS) onboard the European satellite Envisat.

The MLS error budget is reported in Livesey et al. (2011). For HCl observations between 1-20 hPa the precision and accuracy are below 10 and 15% respectively. Between 46 and 100 hPa, these are below 0.3 and 0.2 ppbv, respectively. For $H_2O$ between 0.46 and 100 hPa, precision and accuracy are below 15 and 8%. MIPAS random and systematic errors for various trace gases are reported by Raspollini et al. (2013). For $NO_2$ between 25 and 50 km altitude these are below 10 and 20% respectively. For $HNO_3$ between 15 and 30 km, these are below 8 and 15% while for $O_3$ between 20 and 55 these are below 5 and 10%. At 15 km, these errors increase to 10 and 20%, respectively.

Total column $O_3$ is validated against KNMI's multi sensor reanalysis version 2 (MSR, van der A et al., 2015) which, for the 2008-2010 time period is based on Solar Backscattering Ultraviolet radiometer (SBUV/2), Global Ozone Monitoring Experiment (GOME), SCanning Imaging Absorption spectroMeter for Atmospheric CartograpHY (SCIAMACHY) and Ozone Monitoring Instrument (OMI) observations. The satellite retrieval products used in the MSR are bias-corrected with respect to Brewer and Dobson Spectrophotometers to remove discrepancies between the different satellite data sets. The uncertainty in the product, as quantified by the bias of the observation-minus-analysis statistics, is in general less than 1 DU.

$O_3$ profiles are compared to ozonesonde data that are acquired from the World Ozone and Ultaviolet radiation Data Centre (WOUDC). The precision of the ozonesondes is on the order of 5% in the stratosphere (Hassler et al., 2015), when based on electrochemical concentration cell (ECC) devices (~85% of all soundings). Larger random errors (5-10%) are found for other sonde types, and in the presence of steep gradients and where the ozone amount is low. Sondes at 19, 12, 2 and 1 individual stations are used for the evaluation over northern hemisphere midlatitudes, tropics, southern hemisphere midlatitudes and Antarctic, respectively.

Stratospheric $NO_2$ columns are compared to observational data from the SCIAMACHY (Bovensmann et al., 1999) UV–VIS (ultraviolet–visible) and NIR (near-infrared) sensor onboard the Envisat satellite. The satellite retrievals are based on applying the Differential Optical Absorption Spectroscopy (DOAS) (Platt and Stutz, 2008) method to a 425-450 nm wavelength window. Stratospheric $NO_2$ columns from SCIAMACHY presented here are in fact total columns derived by dividing retrieved slant columns of $NO_2$ by a stratospheric air mass factor and contains data over the clean Pacific ocean (180°E - 220°E) only (Richter et al., 2005). Although in this region the contribution of the troposphere to total column $NO_2$ is small, stratospheric column $NO_2$ from SCIAMACHY is still somewhat positively biased by a tropospheric contribution.

However, stratospheric air mass factors for $NO_2$ are usually large compared to tropospheric ones, so that the uncertainty resulting from this should only have a minor impact on the data analysis presented in this study.

Monthly mean stratospheric $NO_2$ columns are associated with relative uncertainties of roughly 5-10% and an additional absolute uncertainty of $1\times10^{14}$ molec cm$^{-2}$. To account for differences in observation and model output time, simulations are interpolated linearly to the equator crossing time of SCIAMACHY (10:00 LT). In addition, only model data for which satellite observations exist are included in the corresponding comparisons.

Furthermore, satellite-based observations are used from the Atmospheric Chemistry Experiment - Fourier Transform Spectrometer (ACE-FTS), onboard of the Canadian satellite mission SCISAT-1 (first Science Satellite, Bernath et al., 2005). This is a high spectral resolution Fourier transform spectrometer operating with a Michelson interferometer. Vertical profiles of atmospheric volume mixing ratios of trace constituents are retrieved from the occultation spectra, as described in Boone et al. (2005), with a vertical resolution of 3–4 km at maximum. Here we use level 2 retrievals (version 3.0) of $N_2O$ and $CH_4$.

ACE-FTS $N_2O$ observations between 6-30 km agree to within 15% of independent observations, while above they agree to within ±4 ppbv (Strong et al., 2008). The uncertainty in ACE-FTS $CH_4$ observations is within 10% in the upper troposphere – lower stratosphere, and within 25% in the middle and higher stratosphere up to the lower mesosphere (<60 km) (De Mazière et al. 2008).

Three-hourly C-IFS and BASCOE-CTM output has been interpolated in space and time to match with any of these observations.

## 4. Model evaluation

Fig. 2 shows the mean $O_3$ partial columns (PC) against observations from Aura MLS v3.0 over the poles and the tropics. In C-IFS-T, applying the Cariolle parameterization, the annual cycle over the Arctic is very well simulated but a constant overestimation of 50 DU (20%) is found. In the Tropics the bias is much smaller, with a slight underestimation (10 DU, 5%). In the Antarctic, the results are remarkably good during the ozone hole episodes but there is a serious overestimation developing from February until the beginning of August, when it reaches 50 DU (30%) i.e. as much as in the Arctic. CIFS-Atmos and CIFS-TS provide very similar results over the full time period, suggesting that our approach to keep two different solvers in each region is valid for stratospheric ozone. Also after an initialization period of a few months the model runs do not present any obvious drift, and the differences with BASCOE-CTM are very small. This implies that differences due to the model configuration regarding transport are not crucial for lower stratospheric ozone at these timescales. In the Tropics the C-IFS-TS and C-IFS-Atmos results are slightly better than those with BASCOE-CTM, potentially due to the missing parameterization for convection. In the Antarctic, the parameterization of PSC leads to an overestimation of springtime ozone depletion while the Cariolle parameterization simulates very well the lowest columnar values observed in September, as discussed in more detail below. The recovery of ozone is overestimated by 20DU (10%) in December-January. While the amplitude of the annual cycle is overestimated above the Antarctic, its structure matches well the observations.

An evaluation of $O_3$ total columns (TC) against the MSR at various latitude bands is given in Figure S6 in the Supplementary material. Considering the missing tropospheric chemistry in the BASCOE-CTM this system is not well constrained in terms of the $O_3$ TC which implies that it is not useful to include its results here. The TC comparison confirms the evaluation with PC from Aura MLS observations, showing a strong positive bias over the NH mid latitudes and Arctic for C-IFS-T, which is reduced for C-IFS-Atmos and C-IFS-TS. These model versions do not show a significant trend during the 2009 – 2010 period. For the tropical and southern hemisphere mid-latitudes all C-IFS versions show a similar performance with C-IFS-Atmos showing a small positive offset with respect to C-IFS-TS of approx. 2-8 DU depending on the latitude band and season.

Closer inspection of $O_3$ profiles against sondes averaged over the NH-mid latitudes, tropics and SH-mid latitudes for the DJF and JJA seasons in 2009 and 2010 (Figures 3 and 4) shows biases in generally similar order of magnitude, although frequently with opposite sign, for C-IFS-TS and C-IFS-Atmos compared to C-IFS-T. Especially over the extra-tropics the C-IFS-TS and C-IFS-Atmos model versions show lower mixing ratios than C-IFS-T at the middle stratosphere (~10 hPa), generally leading to an improvement compared to the observations. For the NH mid-latitudes this also explains the improved $O_3$ TC and $O_3$ PC in these runs compared to C-IFS-T as discussed above. Nevertheless, these experiments still show a positive bias near the ozone maximum in terms of partial pressure (~50 hPa) and also at lower altitudes during the northern hemispheric spring season. Furthermore, in the tropics the use of the full stratospheric chemistry implies a slight degradation against the linear scheme around the ozone maximum, where the Cariolle parameterization is very well tuned. The negative bias in the lower stratosphere as found in C-IFS-TS is not improved compared to C-IFS-T. These alternating biases in CIFS-TS and C-IFS-Atmos are due to corresponding biases in chemically related species such as $NO_x$ and also to transport issues, as discussed in more detail below. The very similar performance of C-IFS-TS with respect to C-IFS-Atmos, especially in this altitude range, once again gives confidence in our approach to split chemistry scheme for tropospheric or stratospheric conditions. A similar evaluation against MLS observations, but for the period September-October-November 2009, provides very similar conclusions (Figure S7, supplementary material). For the 2009 Antarctic ozone hole season (Fig. 5) the C-IFS-TS and C-IFS-Atmos show a positive bias at ~100 hPa for August and September, and negative bias at higher altitudes (50-10 hPa), where C-IFS-T shows a positive bias. Still the depth of the ozone hole is well modelled in October. During the closure phase in November and December the $O_3$ variability with altitude is better captured in C-IFS-TS than in C-IFS-T.

A closer analysis of the processes responsible for springtime polar ozone depletion is given in Fig. 6. In both the C-IFS-TS and C-IFS-Atmos runs as well as BASCOE-CTM there is an $HNO_3$ deficit at the beginning of the winter. Denitrification, which is not modelled in C-IFS-T, starts at the correct time in the models with stratospheric chemistry indicating that NAT PSC appear at about the right time. However, denitrification proceeds more slowly and ends one month later than observed by Aura-MLS. We attribute this shortcoming to the crude modelling of NAT PSC which does not calculate the amount of condensed nitric acid and water, keeps the surface area densities of PSC particles fixed at an arbitrary value and parameterizes sedimentation through irreversible removal. Chlorine activation starts at exactly the right time and is as strong in the C-IFS runs as in the Aura-MLS observations until the beginning of September, but starts decreasing afterwards while

it lasts two more weeks in the observations. Hence the overestimation of ozone depletion during August and September in the models with explicit stratospheric chemistry is probably not due to an overestimation of chemical removal. This feature is more pronounced in CIFS-TS and CIFS-Atmos than in the BASCOE-CTM, suggesting that it may be associated to differences in the modelling of transport.

The evaluation of the zonal mean ozone mixing ratios against MIPAS observations shows good general agreement, Fig. 7, with all four modelling experiments providing similar features. The tropical maximum of $O_3$ mixing ratio at 10 hPa is under-estimated in all experiments but to a larger extent in those which model stratospheric photochemistry explicitly (BASCOE CTM, C-IFS-TS, C-IFS-Atmos) than in C-IFS-T, in line with the evaluation against $O_3$ sondes for June-July-August (figure 4).  The same evaluation against MLS observations provides exactly the same conclusions (Figure S8, supplementary

material).

The assessment of $NO_2$ against MIPAS daytime $NO_2$ observations, acquired by sampling the ascending orbits from Envisat, shows good agreement with the models, although C-IFS-TS and C-IFS-Atmos tend to show a positive bias. The C-IFS-TS and C-IFS-Atmos runs describe well the seasonal variation in zonal mean stratospheric $NO_2$ columns at different latitude bands, Fig. 8, with monthly mean biases with respect to the SCIAMACHY observations of less than $1 \times 10^{15}$ molec cm$^{-2}$ in

the tropics and at mid-latitudes. The positive bias is larger in C-IFS-Atmos than C-IFS-TS. In contrast, poor performance can be seen for C-IFS-T, due to the lack of stratospheric $NO_x$ chemistry in that version.

However, a positive $NO_2$ bias with respect to night-time MIPAS $NO_2$ observations appears larger for C-IFS-TS and C-IFS-Atmos than for the BASCOE-CTM (Fig. 7). In contrast, this figure also shows a negative bias in $HNO_3$ with respect to MIPAS observations in the BASCOE-CTM, and C-IFS-Atmos, and even more marked in the C-IFS-TS experiment. Even

though a clear improvement compared to run C-IFS-T is found, further investigation is necessary to diagnose the origins of the biases in night-time $NO_2$ above 10 hPa and in $HNO_3$ between 10 and 70 hPa.

Fig. 9 shows an evaluation of $N_2O$ and $CH_4$ profiles during September 2009 against observations by ACE-FTS. Owing to their long lifetimes these trace gases are good markers for the model ability to describe transport processes - i.e. not only the Brewer-Dobson circulation but also isentropic mixing, mixing barriers, descent in the polar vortex, and stratosphere-

troposphere exchange (Shepherd, 2007). Moreover, $N_2O$ is the main source of reactive nitrogen in the stratosphere while $CH_4$ is one of the main precursors for stratospheric water vapour. The figure suggests reasonable profile shapes for both $CH_4$ and $N_2O$ in the upper stratosphere (10 hPa and higher) where their abundance is more strongly influenced by chemical loss but at lower altitudes (100-10 hPa) C-IFS-TS and C-IFS-Atmos show larger discrepancies to the observations than the BASCOE-CTM run, with weaker vertical gradients in the tropics and SH-mid latitudes and a sharper gradient in the extra-

tropical Northern Hemisphere.

This discrepancy cannot be due to different wind fields because the BASCOE-CTM experiment is driven by three-hourly output of the C-IFS experiment. We attribute it instead to the different numerical schemes for advection and/or to differences in the representation of sub-grid transport processes in the GCM and in the CTM. Convection and diffusion are indeed explicitly modelled in C-IFS but neglected in BASCOE CTM, which relies on the implicit diffusion properties of its flux-

form advection scheme to represent sub-grid mixing (Lin and Rood, 1996; Jablonowski and Williamson, 2011). Since lower stratospheric ozone is strongly determined by both chemistry and transport, the transport issue indicated by fig. 9 could also contribute directly to the ozone biases seen below 10 hPa in Figures 3 and 4.

Fig. 10 shows a good consistency between $H_2O$ modelled by C-IFS-TS and the BASCOE-CTM results, albeit with a slight negative bias with respect to MLS observations above 5 hPa, and a positive bias around 30 hPa in the tropics, associated with corresponding biases in $CH_4$. This figure also shows globally a good agreement between HCl modelled by C-IFS-TS and MLS observations, although with a positive bias of 0.8 ppbv confined in the region of ozone depletion above Antarctica.

## 5. Conclusions

We have presented a model description and benchmark evaluation of an extension of the C-IFS system with stratospheric ozone chemistry of the BASCOE model added to the already existing tropospheric scheme CB05. We refer to this system as C-IFS-CB05-BASCOE, or C-IFS-TS in short. In our approach we have retained a separate treatment for tropospheric and stratospheric chemistry, and select the most appropriate scheme depending on the altitude with respect to the tropopause level. This has the advantage that mechanisms which are optimized for tropospheric and stratospheric chemistry, respectively, can be retained, which also substantially reduces the computational costs of the chemical solver compared to an approach where all reactions are activated in the whole atmosphere, referred to as C-IFS-Atmos. Also, it allows for an easy switch between system setups. To avoid jumps in trace gas concentrations at the interface the consistency in gas-phase reaction rates has been verified while the photolysis rates from the two parameterizations are interpolated across the interface. We showed that differences between C-IFS-TS and C-IFS-Atmos are overall small, hence our basic assumption to have different chemistry solvers for troposphere and stratosphere is valid for our applications.

An evaluation of a 2.5 year simulation of C-IFS-TS indicates good performance of the system in terms of stratospheric ozone, of similar quality as its ancestor BASCOE-CTM model results, and a considerable general improvement in terms of stratospheric composition compared to the C-IFS-T predecessor model version which applied a linear ozone scheme in the stratosphere.

The $O_3$ partial columns (10-100 hPa) show biases mostly smaller than ±20 DU when compared to the Aura MLS observations. Also the profiles were generally well captured, and show an improvement with respect to the C-IFS-T linear ozone scheme in the stratosphere over mid-latitudes. The depth and variability of the ozone hole over Antarctica is modelled well. While also the C-IFS-T shows a remarkably good agreement to the observations during the ozone hole episodes it develops a significant overestimation of the partial columns during other months. The tropical maximum of the mixing ratio, around 10 hPa, is the only stratospheric region where C-IFS-T agrees better all-year-long with observations.

Also evaluation of other trace gases ($NO_2$, $HNO_3$, $CH_4$, $N_2O$, HCl) against observations derived from various satellite retrievals (SCIAMACHY, ACE-FTS, MIPAS, MLS) illustrate the clear improvements obtained with C-IFS-TS compared to C-IFS-T, even though C-IFS-TS still suffers from positive biases in stratospheric $NO_2$, whereas $HNO_3$ is biased low. For the

long-lived tracers $CH_4$ and $N_2O$, larger errors with respect to limb-sounding retrievals were found between 10 hPa and 100 hPa than with the BASCOE-CTM, suggesting difficulties in representing slow transport processes. The BASCOE-CTM experiment shown here was driven by three-hourly wind fields output of the C-IFS experiments. Hence this discrepancy is due to a difference in the representation of the transport processes between the GCM and the CTM, i.e. the numerical

scheme used for advection (Semi-Lagrangian versus Flux-form), the convection (parameterized in C-IFS but neglected in BASCOE CTM)  or the diffusion (parameterized in C-IFS but not explicitly considered in the CTM). Hence, stratospheric transport in C-IFS will be an area for further evaluation and developments.

This benchmark model evaluation of C-IFS-TS marks a key step towards merging tropospheric and stratospheric chemistry within IFS, aiming at a possible configuration for daily operational forecasts of lower and middle atmospheric composition

in the near future. Future work could focus on the following aspects:

- Chemical data-assimilation: initial tests with data-assimilation of $O_3$ total column and profile retrievals suggest that stratospheric ozone is successfully constrained in C-IFS-TS. However, observational constraints on other components driving ozone chemistry are currently lacking in the assimilation system. Our extension opens the possibility for assimilation of additional trace gases such as $N_2O$ and HCl. However, for the 4D-VAR assimilation of short-lived species such as $NO_2$

and ClO an adjoint chemistry module would likely be required as implemented the BASCOE DA system.

- Alignment of the reaction mechanism and photolysis rates: while at current stage the gas-phase and photolytic reaction rates of the parent schemes are retained, we foresee a further integration to ensure better alignment of the chemical mechanisms. Especially the existing jumps in photolysis rates as a consequence of the different parameterizations are not desirable, even though they are not harmful for model stability nor visibly lead to any degradation in model performance.

The alignment in terms of gas-phase reaction rate expressions can be achieved by the introduction of the KPP solver in C-IFS, for both tropospheric and stratospheric chemistry, which allows for a better traceable model development than the hard-coded Euler Backward Integration solver as adopted in Flemming et al. (2015).

- Improvement of the representation of stratospheric sulphate aerosols and Polar Stratospheric Clouds: the current climatology for these aerosols, and parameterization for PSCs could easily be improved. While the current results are

satisfactory for a general-purpose monitoring system, these improvements would especially allow better simulations of the composition in in the polar lower stratosphere during springtime.

- Extension of tropospheric and stratospheric chemistry schemes: the availability of a comprehensive set of trace gas fields allows for a relatively easy extension of the tropospheric reaction mechanism by including selective reactions originating from the stratospheric chemistry, and vice versa. Examples are the introduction of halogen chemistry in the troposphere (von

Glasow and Crutzen, 2007), or SO2 conversion to sulphate aerosol in the stratosphere, relevant in case of strong volcanic events (Bândă, et al., 2015).

- Optimization of solver efficiency: even though the use of KPP has simplified the code maintenance and may result in a higher numerical accuracy of the solution, it also caused a considerable slow-down of the numerical efficiency as compared to the Euler Backward Integration solver, as that solver had been optimized for tropospheric ozone chemistry in C-IFS-

CB05. Solutions could be an optimization of the initial chemical time step for the KPP solver, depending on prevailing chemical and physical conditions, and an optimization of the automated solver code, which allows for a more efficient code structure (KP4, Jöckel et al., 2010).

In summary, the extension towards stratospheric chemistry in C-IFS broadens its ability for forecast and assimilation of stratospheric composition, which is beneficial to the monitoring capabilities in CAMS, and may also contribute to advances in meteorological forecasting of the ECMWF IFS model in the future.

**Code availability**

The C-IFS source code is integrated into ECMWF's IFS code, which is available subject to a licence agreement with ECMWF, see also Flemming et al. (2015) for details. The stratospheric chemistry module of C-IFS was originally developed in the framework of BASCOE. Readers interested in the BASCOE code can contact the developers through http://bascoe.oma.be.

# Appendix A

**Table A1.** Trace gases in C-IFS-TS, along with their chemically active domain: troposphere (Trop), stratosphere (Strat) or whole atmosphere (WA).

| Short name | Long name | Active domain |
|---|---|---|
| O3 | ozone | WA |
| OH | hydroxyl radical | WA |
| H2O2 | hydrogen peroxide | WA |
| HO2 | hydroperoxy radical | WA |
| CO | carbon monoxide | WA |
| CH2O | formaldehyde | WA |
| CH3O2 | methylperoxy radical | WA |
| CH3OOH | methylperoxide | WA |
| CH4 | methane | WA |
| NO | nitrogen monoxide | WA |
| NO2 | nitrogen dioxide | WA |
| NO3 | nitrate radical | WA |
| HNO3 | nitric acid | WA |
| HO2NO2 | pernitric acid | WA |
| N2O5 | dinitrogen pentoxide | WA |
| Rn | radon | WA |
| Pb | lead | Trop |

| | | |
|---|---|---|
| C2H4 | ethene | Trop |
| C2H6 | ethane | Trop |
| C2H5OH | ethanol | Trop |
| C3H8 | propane | Trop |
| C3H6 | propene | Trop |
| C5H8 | isoprene | Trop |
| C10H16 | terpenes | Trop |
| CH3COCHO | methylglyoxal | Trop |
| CH3COCH3 | acetone | Trop |
| CH3OH | methanol | Trop |
| HCOOH | formic acid | Trop |
| MCOOH | methacrylic acid | Trop |
| PAR | paraffins | Trop |
| OLE | olefins | Trop |
| ALD2 | aldehydes | Trop |
| ROOH | peroxides | Trop |
| PAN | peroxyacetyl nitrate | Trop |
| ONIT | organic nitrates | Trop |
| SO2 | sulfur dioxide | Trop |
| SO4 | sulfate | Trop |

| | | |
|---|---|---|
| DMS | dimethyl sulfide | Trop |
| MSA | methanesulfonic acid | Trop |
| NO3_A | nitrate | Trop |
| NH2 | amine | Trop |
| NH3 | ammonia | Trop |
| NH4 | ammonium | Trop |
| C2O3 | peroxyacetyl radical | Trop |
| ISPD | methacrolein MVK | Trop |
| ACO2 | acetone product | Trop |
| IC3H7O2 | IC3H7O2 | Trop |
| HYPROPO2 | HYPROPO2 | Trop |
| ROR | Organic ethers | Trop |
| RXPAR | PAR budget corrector | Trop |
| XO2 | NO to NO2 operator | Trop |
| XO2N | NO to alkyl nitrate operator | Trop |
| O | oxygen atom (ground state) | Strat |
| O1D | oxygen atom (first excited) state | Strat |
| H | hydrogen atom | Strat |
| H2 | hydrogen | Strat |
| H2O | Water | Strat |

| | | |
|---|---|---|
| CH3 | methyl radical | Strat |
| CH3O | methoxy radical | Strat |
| HCO | formyl radical | Strat |
| CO2 | carbondioxide | Strat |
| N | nitrogen atom | Strat |
| N2O | nitrous oxide | Strat |
| CL | chlorine atom | Strat |
| CL2 | chlorine | Strat |
| HCL | hydrogen chloride | Strat |
| HOCL | hypochlorous acid | Strat |
| CH3CL | methyl chloride | Strat |
| CH3CCL3 | methyl chloroform | Strat |
| CCL4 | tetrachloromethane | Strat |
| CLONO2 | chlorine_nitrate | Strat |
| CLNO2 | chloro(oxo)azane oxide | Strat |
| CLO | chlorine monoxide | Strat |
| OCLO | chlorine dioxide | Strat |
| CLOO | asymmetric chlorine dioxide radical | Strat |
| CL2O2 | dichlorine_dioxide | Strat |
| BR | bromine atom | Strat |

| | | |
|---|---|---|
| BR2 | bromine atomic ground state | Strat |
| CH3BR | methyl bromide | Strat |
| CH2BR2 | dibromomethane | Strat |
| CHBR3 | bromoform | Strat |
| BRONO2 | bromine nitrate | Strat |
| BRO | bromine monoxide | Strat |
| HBR | hydrogen bromide | Strat |
| HOBR | hypobromous acid | Strat |
| BRCL | bromine monochloride | Strat |
| HF | hydrofluoric acid | Strat |
| CFC11 | trichlorofluoromethane | Strat |
| CFC12 | dichlorodifluoromethane | Strat |
| CFC113 | trichlorotrifluoroethane | Strat |
| CFC114 | 1,2-dichlorotetrafluoroethane | Strat |
| CFC115 | chloropentafluoroethane | Strat |
| HCFC22 | chlorodifluoromethane | Strat |
| HA1301 | bromotrifluoromethane | Strat |
| HA1211 | bromochlorodifluoromethane | Strat |

**Acknowledgments**

MACC III was funded by the European Union's Seventh Framework Programme (FP7) under Grant Agreement no. 283576. We are grateful to the World Ozone and Ultraviolet Radiation Data Centre (WOUDC) for providing ozone sonde observations and to the GOME-2, MIPAS, ACE-FTS and MLS teams for providing satellite observations.

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

**Table 1.** Trace gases relevant for the stratosphere which are constrained at the surface. The constant surface volume mixing ratios are also given.

| $N_2O$ | CFC11 | CFC12 | CFC113 | CFC114 | $CCl_4$ | $CH_3CCl_3$ |
|--------|-------|-------|--------|--------|---------|-------------|
| 3.22E-7 | 2.59E-10 | 5.37E-10 | 7.93E-11 | 4.25E-12 | 1.02E-10 | 4.53E-11 |
| HCFC22 | HA1301 | HA1211 | $CH_3Br$ | $CHBR_3$ | $CH_3Cl$ | $CO_2$ |
| 1.70E-10 | 3.30E-12 | 4.62E-12 | 9.08E-12 | 1.17E-12 | 5.44E-10 | 3.80E-4 |

5 **Table 2.** Number of trace gases, the chemistry scheme in troposphere and stratosphere, and corresponding number of reactions (gas-phase / heterogeneous and photolytic), as well as specification of the circulation model and computational expenses of a one-month run on T255L60 in terms of system billing units (SBU) for various C-IFS model versions. For completeness also the BASCOE-CTM system is indicated.

| | C-IFS-T | C-IFS-S | C-IFS-Atmos | C-IFS-TS | BASCOE-CTM |
|---|---------|---------|-------------|----------|------------|
| No. trace gases | 55 | 59 | 99 | 99 | 59 |
| Chemistry scheme in troposphere | CB05 | BASCOE (P<400hPa) | CB05+BASCOE | CB05 | BASCOE (P<400hPa) |
| Chemistry scheme in stratosphere | CB05/Cariolle | BASCOE | CB05+BASCOE | BASCOE | BASCOE |
| No. reactions (gas / het / photo) | 93/3/18 | 142/9/52 | 211/11/60 | 93/3/18 *or* 142/9/52 | 142/9/52 |
| Circulation model | GCM | GCM | GCM | GCM | CTM |
| SBU | 2075 | 2500 | 4563 | 3076 | - [a] |

[a]BASCOE does not run on the ECMWF supercomputing facility and hence cannot be compared directly to C-IFS in terms of
10 computational resources.

**Table 3.** Parameterization of photolysis rates for troposphere (CB05-based) and stratosphere (BASCOE-based)

|  | Troposphere (Williams et al., 2012) | Stratosphere (Errera and Fonteyn, 2001) |
|---|---|---|
| **No. J-rates** | 18 | 52 |
| **Method** | 2-stream online solver, $204<\lambda<705nm$ | Lookup table approach, $116<\lambda<705nm$ |
| **Dependencies** | $O_3$ overhead, pressure, solar zenith angle, cloud, aerosol, surface albedo, temperature | $O_3$ overhead, pressure, solar zenith angle |
| **terminator treatment** | J>0 for sza<85° | J>0 for sza<96°, Chapman approximation |

**Table 4.** Selection of photolytic reactions that are merged between troposphere and stratosphere. The reaction product $O_2$ is not shown.

| Name | reaction (stratosphere) | reaction products (troposphere)[a] |
|---|---|---|
| J O3 | $O_3 + hv \rightarrow O^1D$ | |
| J NO2 | $NO_2 + hv \rightarrow NO + O$ | $NO + O_3$ |
| J H2O2 | $H_2O_2 + hv \rightarrow 2OH$ | |
| J HNO3 | $HNO_3 + hv \rightarrow OH + NO_2$ | |
| J HO2NO2 | $HO_2NO_2 + hv \rightarrow HO_2 + NO_2$ | |
| J N2O5 | $N_2O_5 + hv \rightarrow NO_2 + NO_3$ | |
| J CH2O-a | $CH_2O + hv \rightarrow HCO + H$ | $CO + 2HO_2$ |
| JCH2O-b | $CH_2O + hv \rightarrow CO + H_2$ | $CO$ |
| J NO3-a | $NO_3 + hv \rightarrow NO_2 + O$ | $NO_2 + O_3$ |
| J NO3-b | $NO_3 + hv \rightarrow NO$ | |
| J O2 | $O_2 + hv \rightarrow 2O$ | |
| J CH3OOH | $CH_3OOH + hv \rightarrow CH_3O + OH$ | $CH_2O + HO_2 + OH$ |

[a] Only specified in case this is different from the stratospheric reaction.

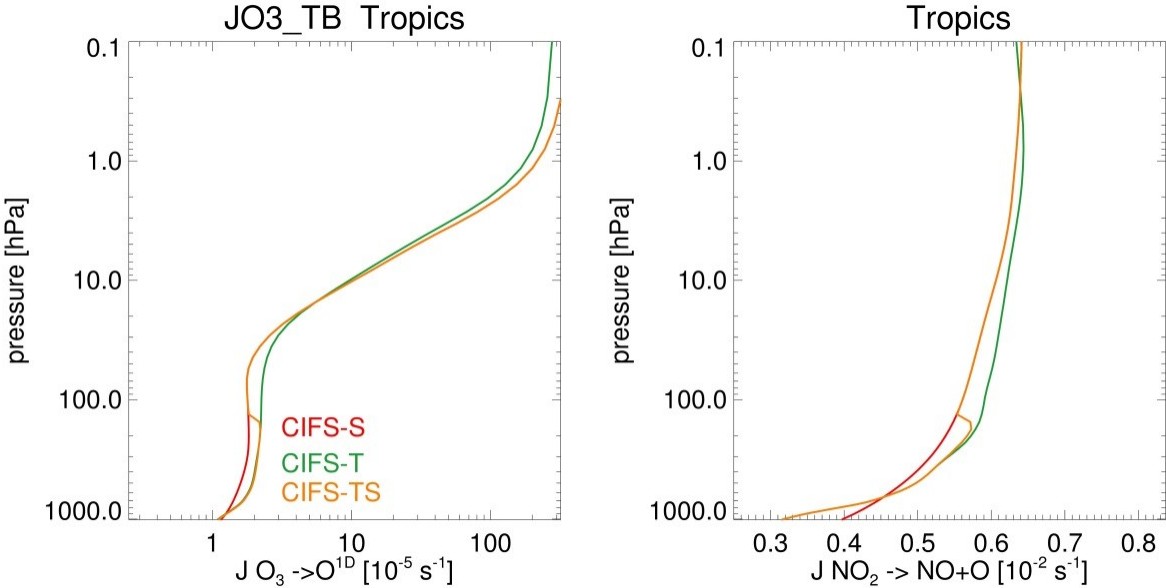

**Figure 1.** Illustration of the merging procedure for photolysis rates between the tropospheric and stratospheric parameterizations for the reaction $O_3 \rightarrow O^1D$ (left) and $NO_2 \rightarrow NO+O$ (right) as zonally averaged over the tropics for 1 April 2008.

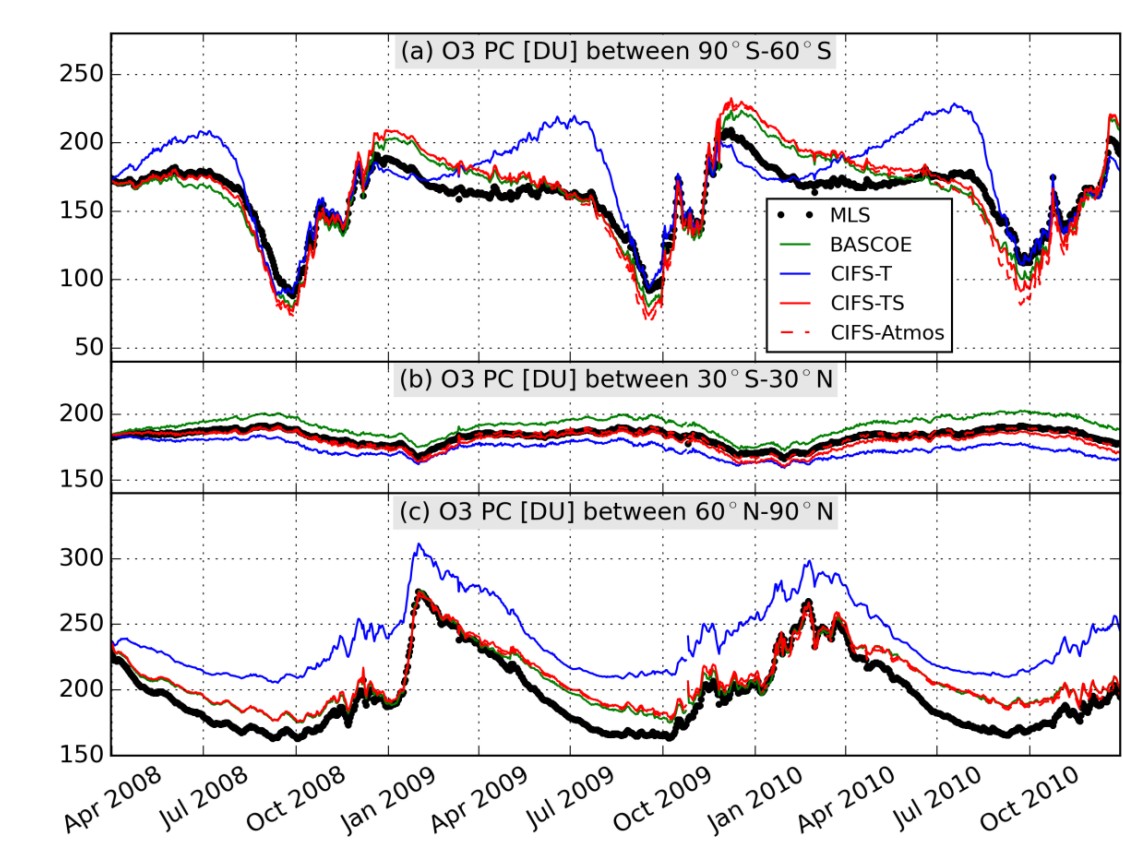

**Figure 2.** Daily averages of $O_3$ partial columns (10-100hPa) for the Arctic (60°N-90°N), Tropics (30°S-30°N) and Antarctic (60°N-90°N) over the period April 2008 – December 2010. Datasets are averaged in 5-day bins and model output is interpolated to the location and time of Aura MLS v3 retrievals (black dots). Blue line: C-IFS-T; green line: BASCOE-CTM; red dashed line: C-IFS-Atmos; red solid line: C-IFS-TS.

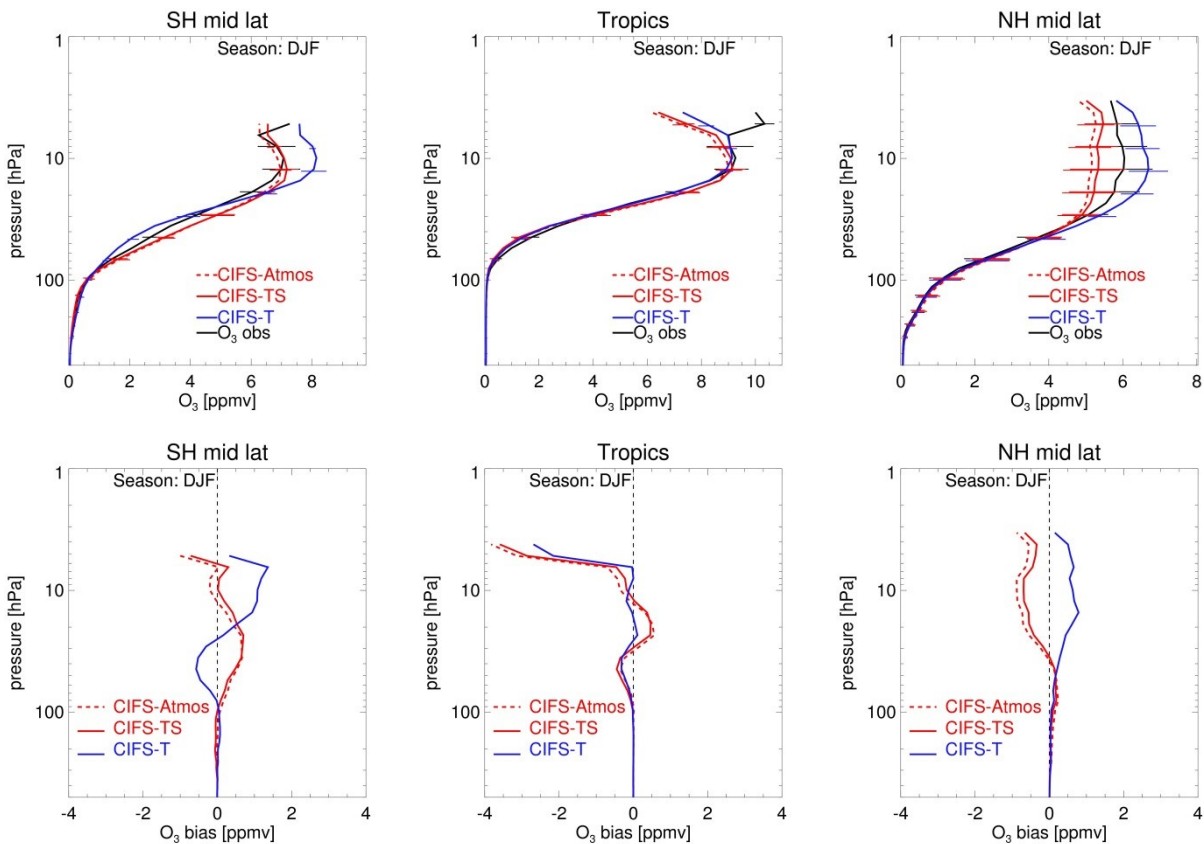

**Figure 3.** Top row: evaluation of ozone against WOUDC sondes over SH mid-latitudes (60°S-30°S, left), tropics (30°N-30°S, middle) and
5    NH mid-latitudes(30°N-60°N, right) for December-January-February 2009 and 2010 in units ppmv. Black: WOUDC observations, red
dashed: C-IFS-Atmos, red solid: C-IFS-TS, blue: C-IFS-T. Error bars denote the 1-sigma spread in the models and observations. Bottom
row: corresponding mean biases.

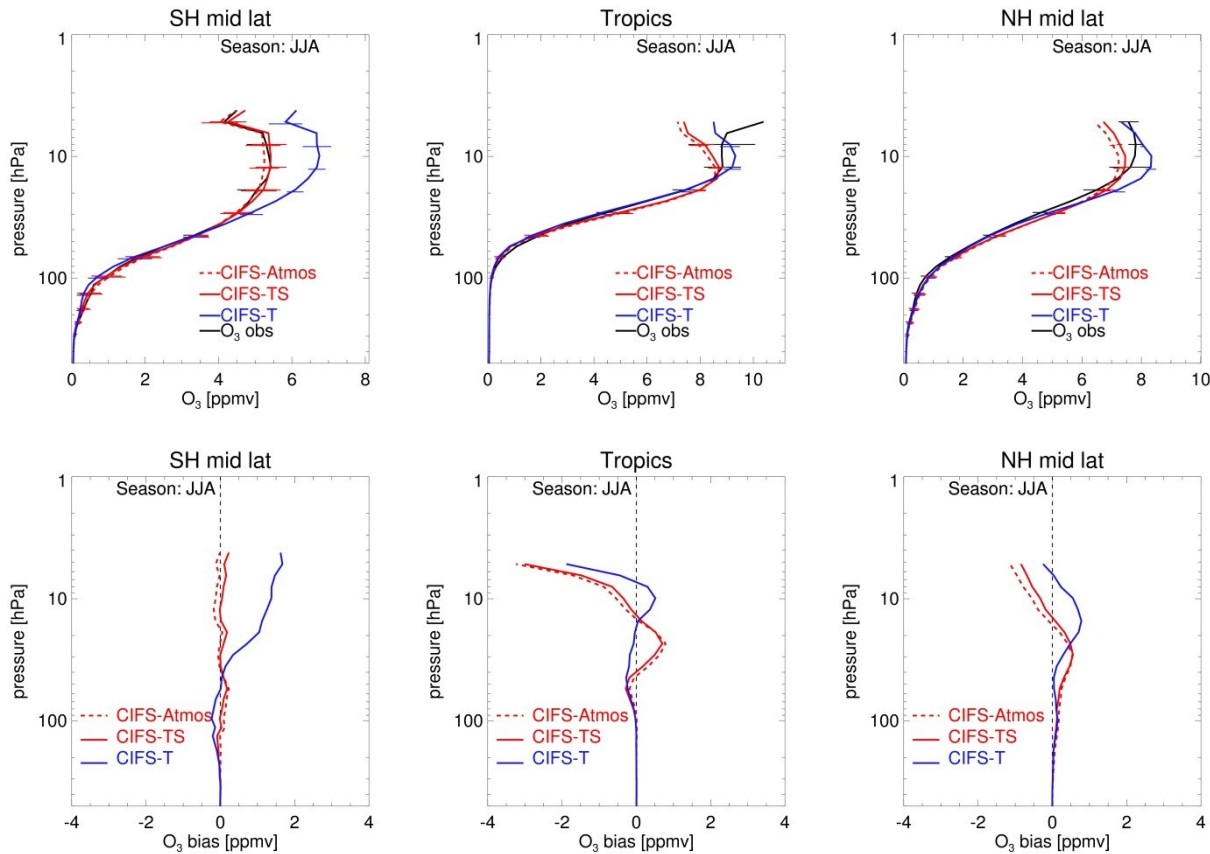

**Figure 4.** Same as Fig. 3, but for June-July-August 2009 and 2010.

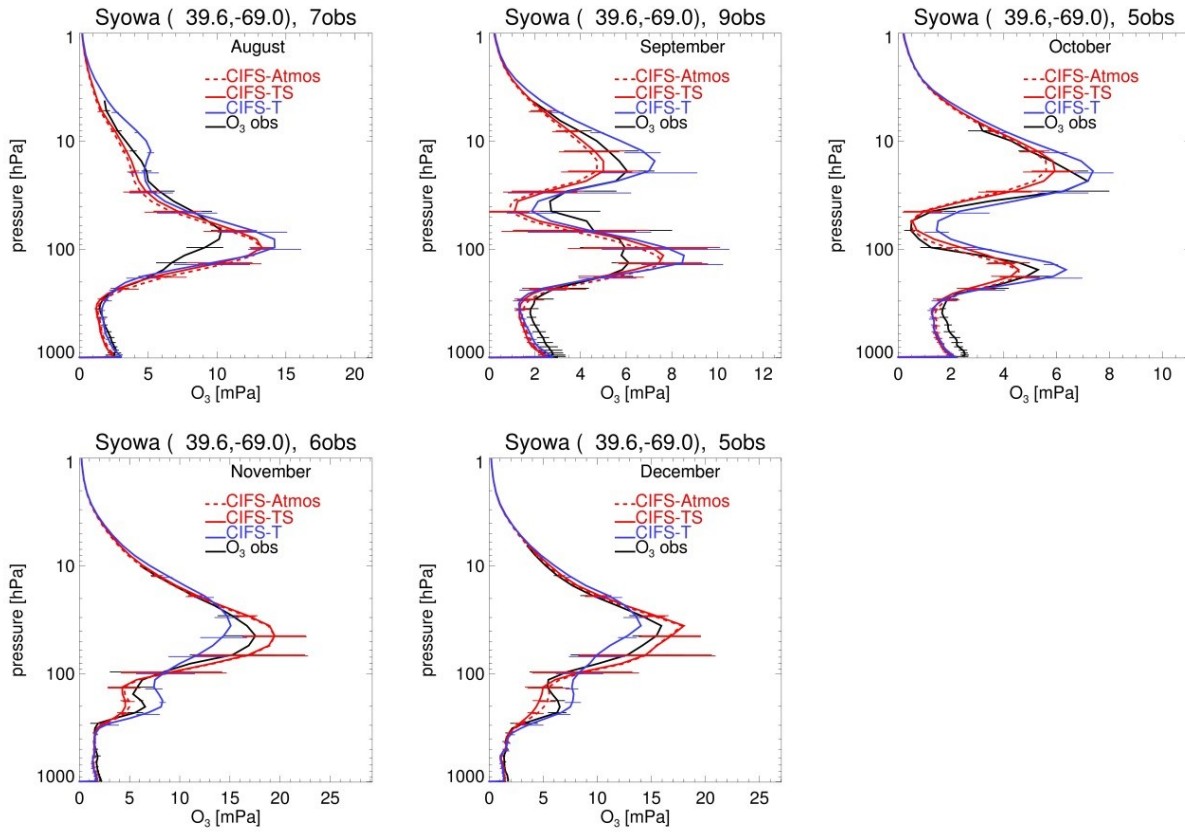

**Figure 5.** Evaluation of ozone in units mPa against WOUDC ozone sondes at Syowa station during August-December 2009. Black: ozone sonde, red dashed: C-IFS-Atmos, red solid: C-IFS-TS, blue: C-IFS-T. Error bars denote the 1-sigma spread in the models and observations.

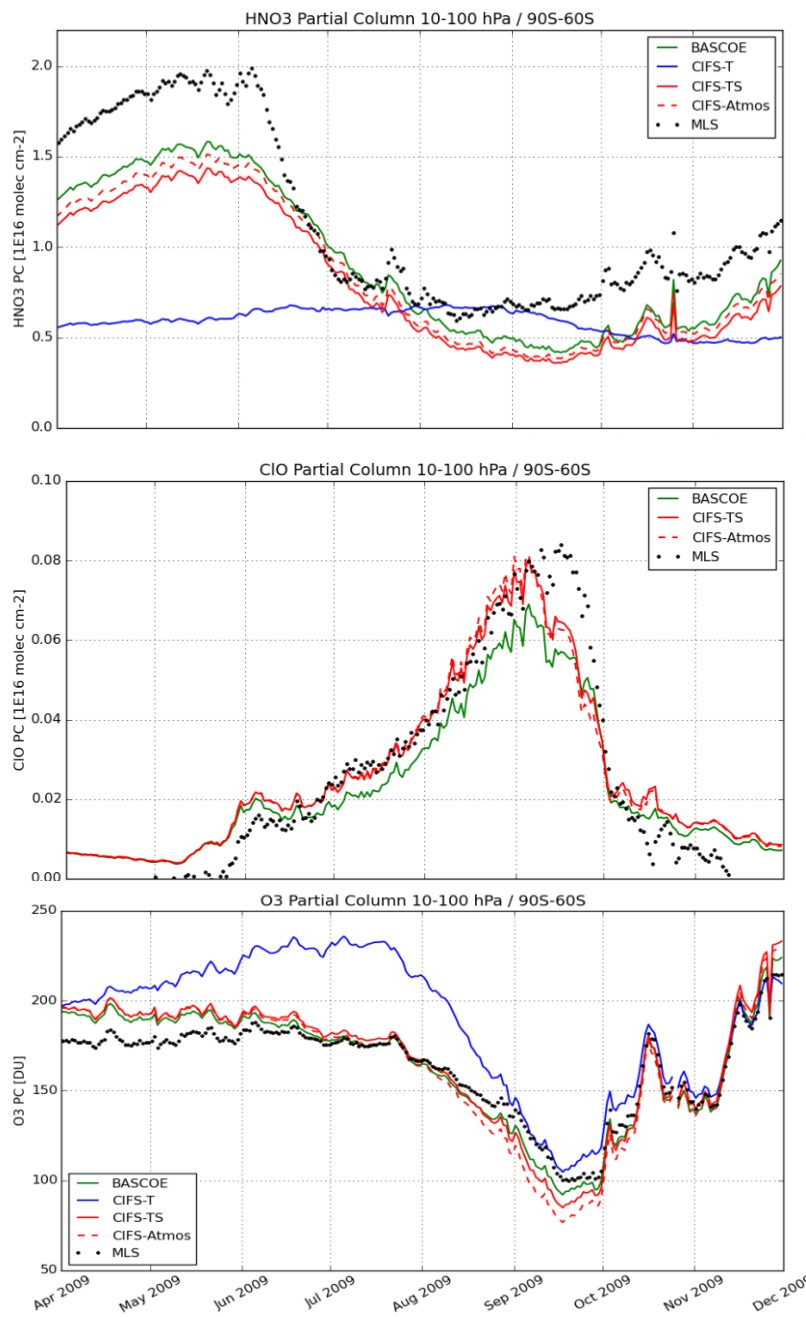

**Figure 6.** Daily averages of $O_3$ partial columns (10-100hPa) over the Antarctic (90°S-60°S), for the period April – November 2009 for $HNO_3$ (top), ClO (middle) and $O_3$ (bottom) against MLS observations.

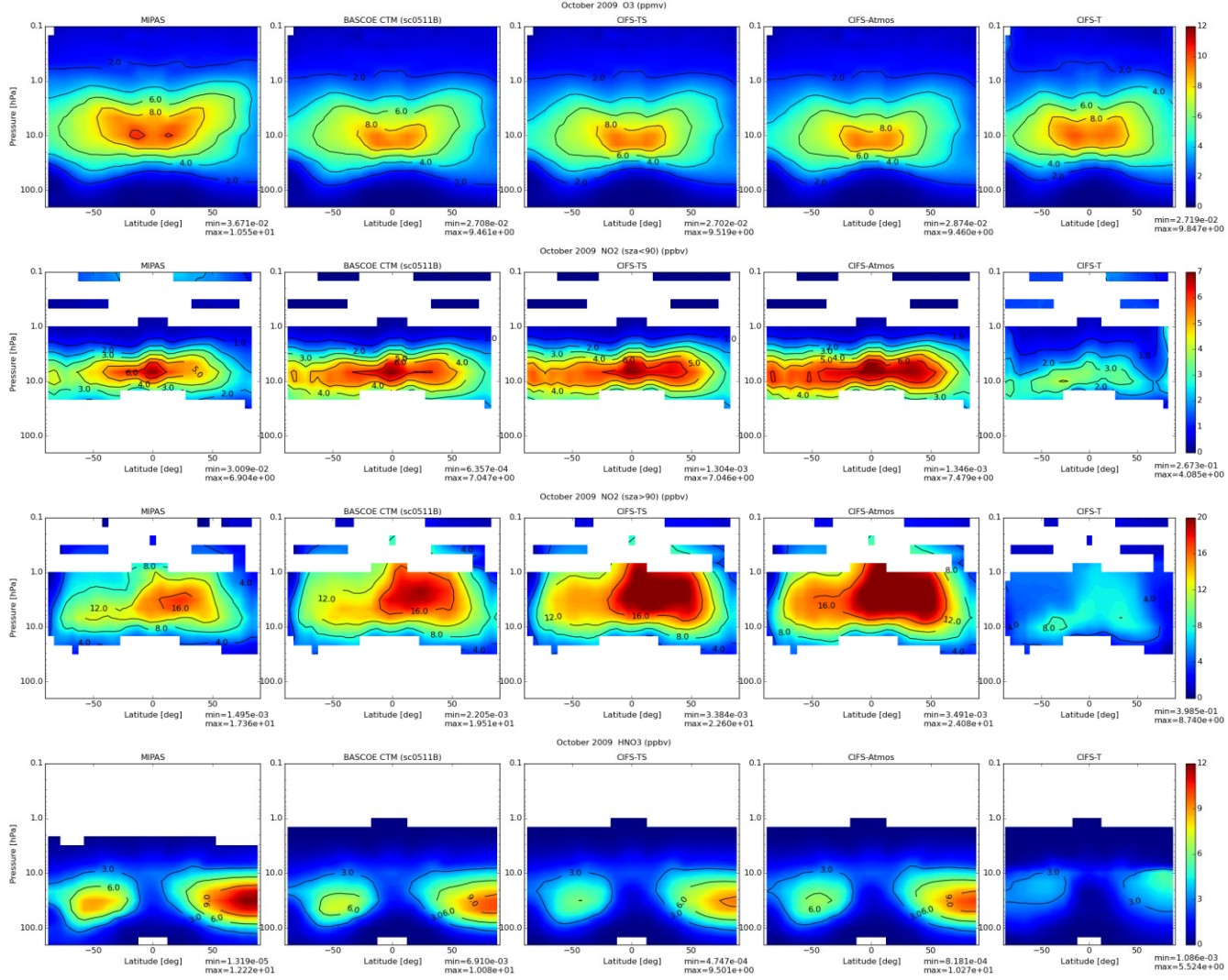

**Figure 7.** Zonal mean stratospheric O$_3$ (top row, units ppmv), daytime NO$_2$ (second row), night-time NO$_2$ (third row) and HNO$_3$ (bottom row, all in units ppbv) for October 2009 using MIPAS observations (first column) and co-located output of BASCOE-CTM (second),C-IFS-TS (third), C-IFS-Atmos (fourth) and C-IFS-T (fifth).

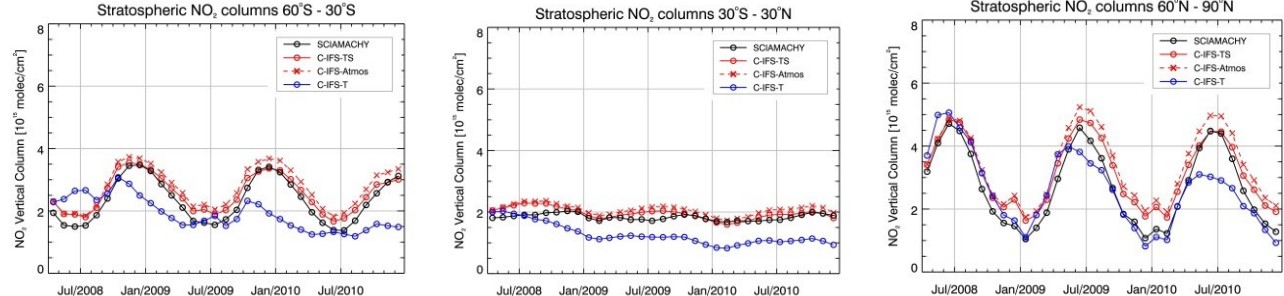

**Figure 8.** Time series of total column $NO_2$ above the clean Pacific ocean (180°E-220°E) for April 2008 – Dec 2010, in units $10^{15}$ molec cm$^{-2}$ for NH mid-latitudes (left), tropics (middle) and SH mid-latitudes (right). Black: SCIAMACHY observations, red dashed: C-IFS-Atmos, red solid: C-IFS-TS, blue: C-IFS-T.

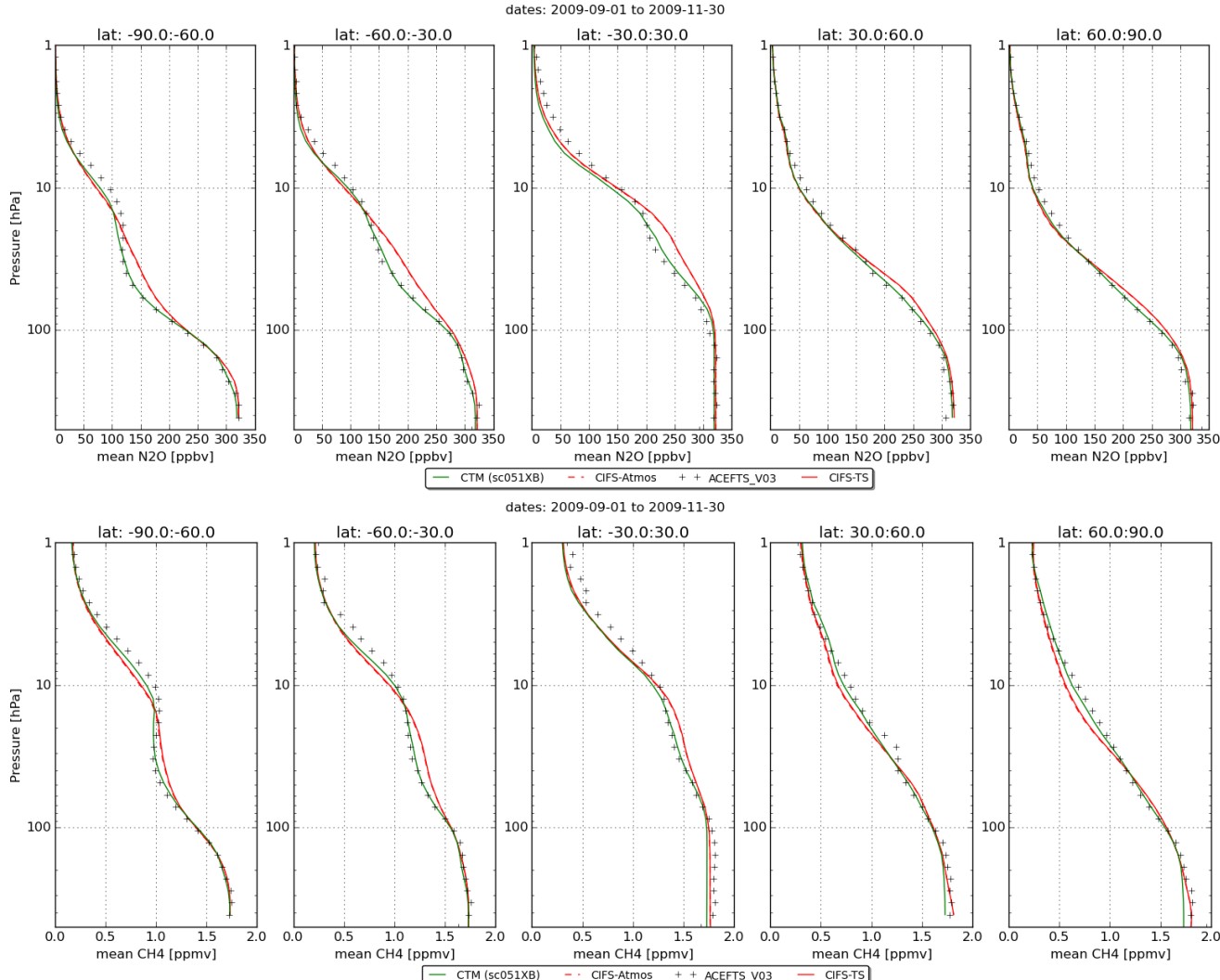

5   **Figure 9.** Zonal mean profiles of stratospheric $N_2O$ (top) and $CH_4$ (bottom) for September-October-November 2009 using ACE-FTS observations (black symbols) and co-located output of BASCOE-CTM (green lines), C-IFS-TS (red solid lines) and C-IFS-Atmos (red dashed lines). The zonal means are shown separately on five columns corresponding to the latitude bands 90°S-60°S, 60°S-30°S, 30°S-30°N, 30°N-60°N and 60°N-90°N, respectively.

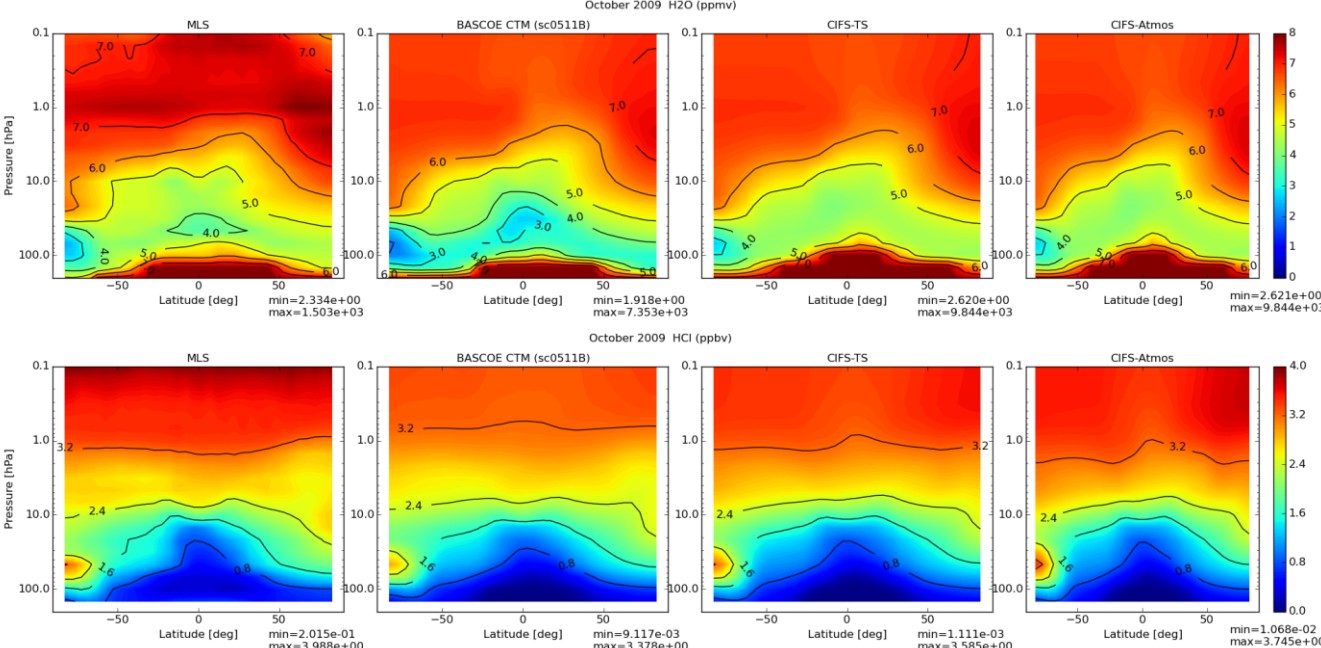

**Figure 10.** Zonal mean stratospheric H$_2$O (top, units ppmv) and HCl (bottom, units ppbv) for October 2009 using Aura/MLS observations (first column) and co-located output of BASCOE-CTM (second), C-IFS-TS (third) and C-IFS-Atmos (fourth).

