# Peer review of "C-IFS-CB05-BASCOE: Stratospheric Chemistry in the Integrated Forecasting System of ECMWF"

_Geoscientific Model Development, 2016_

## Referee Comment (RC1) · Anonymous Referee #1 · 29 Mar 2016

The paper describes an update of the C-IFS model, including a stratospheric chemistry scheme in addition to the existing tropospheric scheme, and using the tropopause to switch between the schemes. This method is also used by other models, and the authors consider the possible inconsistency at the transition zone between the domains.

The paper is fairly short and concise, although there are a few issues that should be improved or made clearer.

I therefore think minor revisions are necessary, although my questions on the relatively short simulation period may require more effort than the other comments.

[Figure]

**Abstract**

The abstract should contain some more on the motivation for including stratospheric chemistry in CAMS. Is forecasting skill part of the motivation?

It is stated that this is a first step, which makes me wonder how far away the next step really is.

**1. Introduction**

Page 1, Line 25: "Also analyses and forecasts of stratospheric ozone directly impact the forecast capabilities of surface solar irradiance" is a bit difficult to understand. Suggest changing to e.g. "Also, the amount of stratospheric ozone directly impact the forecast capabilities of surface solar irradiance, making good stratospheric ozone forecast important".

Page 3, Line 5: "We have developed a strategy": Perhaps better to say "We present here a merging ...". There is no need to state that there is a strategy, and the work done is the actual merging.

**2.1 Stratospheric chemistry**

Page 4, Line 15: Are the surface area densities (SAD) fixed? No size distribution?

On the fact that you do not do sedimentation, but parameterise it using exponential decay of $HNO_3$ and $H_2O$, is this a good approximation?

If you have a situation of very cold temperatures over a long period, the PSCs may sediment out and the SAD will be reduced. Will the parameterisation cause too large

denitrification/dehydration? Figure 9 could perhaps indicate this?

**2.3 Merging procedure ...**

Page 5, Line 24: 40 hPa as tropopause is very low. Does this occur often? To my knowledge, the tropopause pressure is seldom lower than 80hPa.

Using 40 hPa may be of small consequence, and may even cause O3 production to be better represented.

Page 5, Line 26: "Specificaly" -> "Specifically".

Page 5, Line 34: 10-day decay rate in the stratosphere. What happens to the lost species? Are they assumed to be converted to aerosols?

Page 6, Line 11-14: It would be very interesting to see how chemical composition changes when compared to this test. I think some more info should be given on this, as it could explain chemically why you use two domains.

**2.3.1 Merging photolysis rates**

Page 6, Line 26-27: How is the interpolation/merging done? Some weighting for different layers?

Page 7, Line 1: The merging only affects 4 layers, where some are in the troposphere, so I am not surprised the over-all impact is small. Perhaps it could be noted that JO3 does not seem to be used in troposphere?

Did you check hourly composition at these grid boxes?

**2.3.2 Merging tracer transport**

This is perhaps not really a merging.

Regarding stratospheric $H_2O$ tracer: So you do not use tropospheric $H_2O$ (from $q$) as source of stratospheric H2O? I would expect some boundary condition is needed, at least in the tropics. How is this treated?

**3.1 Observational data ...**

Page 8, Line 21: "weighted part": What kind of weight?

Page 8, Line 23: What is the model output frequency? It would be helpful to specify this and whether you use instant model values.

**4 Model evaluation**

Page 9, Line 14-20: The equatorial low bias and high bias at NH mid-lat could indicate too fast transport away from Equator?

Figure 2: The simulated time period is not long enough to make clear whether the bias will build up. This should be further explained.

Ozone hole: Do you overestimate PSCs (as commented earlier) and hence halogen activation?

Page 9, Line 20-32: Is the comparison done by extracting the same profiles (location and time) as in sondes? Should be specified.

The sonde comparison should include some variability in sondes (e.g. as standard

deviation), and possibly also from the model (Fig.3-5). (In Figure 3-4 this could be placed in either upper or lower row.)

Figure 6: How is OH in IFS vs BASCOE? Do you have less HNO3 in stratosphere because of hotter photochemistry? Some thoughts/explanation should be given for the higher low-SZA NO2 in IFS than in BASCOE. NO2+NO3 <-> N2O5 at night? Photochemistry?

Page 10, Line 3-4 (Figure 7): NH higher NO2: Could a possible reason be that SCIA-MACHY assumes that a too high fraction of the column is located in the troposphere?

Page 10, Line 19: What about too fast horizontal transport? Is CH4 fixed at surface?

**5 Conclusions**

Page 11, Line 3: I generally do not think 1.5 years is enough for evaluating the chemistry and chemistry. A possible drift in the $O_3$ column (Fig. 2) should be investigated. If chemistry is adjusted in an assimilation system, a drift will probably not be very prominent or important, but used as a CTM 1.5 years is short.

Page 11, Line 9-11: "a larger error" -> "larger errors", "was" -> "were", and fix rest of sentence.

Page 11, Line 12: Why is it necessary to do this first step? It seems another step is expected.

Page 11, Line 18: What use would assimilation of long-lived species have in the IFS?

Page 12, Line 6: While monitoring capabilities may be important and interesting in the stratosphere, and also provide global products of species which are not globally observed, it would be interesting to hear the implications for forecasting.

Is it not only O3 that is important for radiation calculations? Better stratospheric O3 could improve stratospheric temperatures?

**Appendix**

Table A1 should be sorted on names.

**References**

DOIs are missing for some references; please update.

---

## Short Comment (SC1) · 30 Mar 2016

Dear authors,

In my role as Executive editor of GMD, I would like to bring to your attention our Editorial version 1.1:

http://www.geosci-model-dev.net/8/3487/2015/gmd-8-3487-2015.html

This highlights some requirements of papers published in GMD, which is also available on the GMD website in the 'Manuscript Types' section:

http://www.geoscientific-model-development.net/submission/manuscript_types.html

[Figure]

In particular, please note that for your paper, the following requirement has not been met in the Discussions paper:

- "The main paper must give the model name and version number (or other unique identifier) in the title."

Please add a version number or unique identifier either for each of the models you combined or for your newly created model in the title upon your revised submission to GMD.

Yours,

Astrid Kerkweg

––––––––––––––––––––––––––––––––

---

## Referee Comment (RC2) · Anonymous Referee #2 · 26 Apr 2016

The purpose of the paper is to describe and benchmark a new version of the IFS model. This version has separate chemistry modules (mechanisms) for the troposphere and stratosphere, where the decision of which module to call is determined by the altitude of the grid box with respect to the tropopause. The stratospheric chemical mechanism comes from an assimilation system (BASCOE). The previous version of the IFS had tropospheric chemistry plus linearized stratospheric O3. The paper concludes that a new simulation that uses both chemical mechanisms (called CIFS-TS) has good stratospheric O3, NO2 and other reactive trace gases compared to satellite data sets.

A goal of the paper is to demonstrate that their method of using the tropospheric mechanism/solver for tropospheric grid boxes and the stratospheric solver for stratospheric

grid boxes is a computationally efficient way to calculate the full chemistry of the atmosphere. The biggest problem with this paper is that they did not actually test this. To demonstrate their method, they need to have run a simulation where tropospheric and stratospheric reactions were solved TOGETHER and NOT split into 2 mechanisms. Those results could then be compared with their 'split' method. Ideally, this would show that their method was faster (how much faster?) yet produced essentially the same results. I recommend they do this and then rewrite this manuscript. This experiment would not only satisfy the stated goal of the paper but it would also eliminate the confusion in the comparisons (see below) regarding transport/advection differences between BASCOE-CTM and the CIFS-TS.

I don't agree with the statement in the abstract that the new model configuration shows good performances of stratospheric O3, NO2, and other tracers. The figures chosen to demonstrate good representation of various stratospheric constituents in the CIFS-TS model generally show fair to poor agreement with observations. Stratospheric O3, for example, often looks worse (or at least no better) that it did in the CB05 (trop only) or BASCOE (strat only) versions. This new model does not appear to be an improvement over previous model versions.

The comparisons between CIFS-TS and BASCOE-CTM are confusing. When stratospheric species such as NO2, HNO3, and O3 are compared, the results are different. I thought the primary goal of the paper was to compare the chemical mechanisms, but since the results are rather different, there must be transport (or meteorological field) differences too. This is alluded to on page 7, lines 26-28. The transport/advection needs to be the same in the two simulations in order to compare the chemical mechanisms. It should be made clearer in the text what the differences are between CIFS-TS and BASCOE-CTM.

My overall recommendation is to test a combined (strat+trop) solver and compare the results to trop only, strat only, and the 'split' method presented here. The results will provide a good benchmark and will be easier to interpret if all experiments are performed with the same transport code and meteorological fields.

Other points

It would be helpful to add a table that lists the specifications of each of the models used and notes how dynamical fields are obtained (e.g., forecast, assimilation, . . .?), chemical mechanism, resolution, etc. For example, BASCOE is an assimilation system, but it's only the BASCOE stratospheric chemical mechanism that used here, right? And BASCOE-CTM means the assimilated (renanalysis) fields have been saved and then are being used in an offline chemistry transport model? Presumably it is the same offline model that the C-IFS forecast fields are used in? If what I am asking does not make sense, please take this as an indication that I am confused by the descriptions of the models.

Regarding 'tracer species' or similar expression found in many places, 'tracer' means a species that is unreactive and can be used to trace something, like transport. I think you mean 'trace gas' rather than tracer because that can be used in a general way to talk about any type of constituent in the model. Please search on 'tracer' in the document to identify where you mean trace gas or constituent.

p. 3, l.24. Are you saying the chemistry in the modules is parameterized? Or are you referring to the chemical mechanisms when you say 'chemical parameterization'? A parameterization for chemistry is not the same thing as a chemical mechanism. Some times 'chemical schemes' is used, which is fine for referring to the mechanism. This confusion occurs throughout the paper. Please check each occurrence of 'parameterization' to verify the right words were chosen.

p. 4, l. 15. The threshold temperature for NAT formation is pressure dependent. The manuscript indicates that 194 K was chosen as the threshold regardless of pressure. That would not be the correct way to calculate it.

p. 5, l. 31. I don't understand what is meant by O1D and O3P being described implicitly,

**[GMDD](** )

Interactive
comment

as opposed to being treated explicitly.

p. 6, l. 24. 'solar radiation reaches the stratosphere earlier than the surface...' as written this sounds like it is referring to delay caused by the speed of light! I doubt this was intended; it needs better wording.

p. 7, Section 2.3.1. For JO3, the lack of a 'jump' in O3 may be because photolysis is unimportant (slow) near 100 hPa, so O3 is probably long-lived relative to the photochemical lifetime. JNO2 is much larger so I'm not sure why there isn't a jump – can you explain this? It would be useful if you showed the simulated O3 and NO2 profiles in Fig. 1 to demonstrate the lack of a jump. What is the meaning of 'JO3_TB' in the title of one plot? No similar title for the other plot.

Section 2.3.2, l. 8. It's unclear whether you're saying NO, NO2, and H2O have the mass fixed applied or whether they are the few species where the mass fixed isn't applied. How badly is H2O not conserved in the stratosphere? This will conceivably cause problems for stratospheric chemistry. It would be useful to see a 1-year time series of the H2O mass above 100 hPa.

p. 7, last 3 lines. This sentence says you are looking to identify differences in transport schemes. This confuses the issue of evaluating the chemical mechanisms (and their implementation). This evaluation should be performed using the same dynamical fields with the same model. If the advection schemes are also different, then we cannot actually test the impact of chemical mechanisms alone. And does 'parameterization' in line 28 refer to the different chemical mechanisms?

p. 8, l. 27, 'first Science Satellite'?

p. 8, l. 30-31. Suggest to change to '...between 6-30 km agree to within 15% of independent ...'' For all the figures that are line plots (starting with Figure 2), the blue and black lines are hard to distinguish. Please do something with the line thickness and colors to improve readability.
Section 4, Model Evaluation p. 9, lines 14-19. This paragraph would benefit by a general statement of the purpose of this comparison. It appears the purpose is to show that the TS mechanism looks more like the observed total column O3 than does the trop-only code (with linearized strat O3). One would expect the TS O3 to be better than the linearized O3 of CB05, but there should also be a comparison with the strat-only code. Comparing with the O3 results in Fig. 6, I think the strat chem O3 columns would be lower than the TS mechanism. I guess they aren't the same because the BASCOE-CTM has different transport. Again, not having the same transport in all the simulations really interferes with a useful comparison.

p. 9, discussion of Figs. 3-4. I do not agree that there are meaningful, reduced biases in the TS version. The linearized O3 chemistry of the trop mechanism gives different results from the TS version, but not really worse. These figures show that TS is not an improvement over trop-only. I think the use of mPa for the O3 bias (lower panels) is misleading and probably minimizes the appearance of the disagreement in the middle stratosphere.

p. 9, discussion of Fig. 5. I cannot tell the difference between obs and CIFS-T lines in the figure. There is no line color/style for the observations in each panel's legend. The TS O3 agrees with one of the black lines (obs or CIFS-T??) near and below 100 hPa – sometimes – but the TS O3 consistently has poor agreement above 50 hPa. Why? Since the TS (red) line often does not agree with either black line – I see no basis for claiming good agreement. Additionally, Syowa is often near the vortex and has large daily variability. Were the simulated profiles used in this figure calculated from the same days of the month as the Syowa data?

p. 9, lines 31-32. If you made a difference plot between MIPAS and the simulations, then you might be able to say whether there is good agreement. As presented, the conclusion can't be drawn that there are 'small biases'. Near the tropical maximum the TS looks slightly better than the BASCOE-CTM. Again, assuming that some of the differences are due to dynamical fields or advection scheme, this comparison isn't very

useful.

p.10, lines 5-9. What is the message here? The CIFS has a terrible high bias in nighttime NO2 and a large low bias in HNO3. Why is the CIFS simulation worse than BASCOE? There is no clear explanation here.

p. 10, lines 10-20 (Fig. 8). N2O and CH4 profiles do NOT assess vertical transport. Their profiles below ~10 hPa represent a balance between the vertical and horizontal components of the residual mean circulation. That balance depends on latitude, that is, whether the profile is from the tropical upwelling region or somewhere in the midlatitudes (horizontal and vertical motions matter and so does mixing), or isolated inside the polar vortex (descent). Above 10 hPa, profiles are more strongly influenced by chemical loss so the 2 simulations should look very similar there. The CIFS-TS simulation tends to look worse than the BASCOE CTM or the observations between 10-50 hPa. This suggests circulation and/or mixing problems in the tropics and SH. O3 at 20 hPa is strongly influence by chemistry, not just transport. These paragraphs indicate a lack of understanding of transport circulation and its diagnosis, as well as any understanding of what controls stratospheric ozone distributions.

————————————————————

---

## Author Comment (AC1) · 8 Jul 2016

**Response to the reviewer comments on the manuscript**

**C-IFS-CB05-BASCOE: Stratospheric Chemistry in the Integrated Forecasting System of ECMWF**

By V. Huijnen et al.

First, we would like to thank the reviewers for their critical, but useful comments. In view of their valuable suggestions in our revised manuscript we have:

1   included an additional model configuration containing full (tropospheric *and* stratospheric) chemistry within the whole atmosphere
2   revised our PSC-parameterization
3   extended our model evaluation with one additional year
4   revised some of our evaluations

The reviewer's comments are given in italic, and our responses in regular font. Textual modifications to the manuscript are highlighted in bold. Figure numbers refer to the revised manuscript.

*Response to anonymous Referee #1*

*The paper describes an update of the C-IFS model, including a stratospheric chemistry scheme in addition to the existing tropospheric scheme, and using the tropopause to switch between the schemes. This method is also used by other models, and the authors consider the possible inconsistency at the transition zone between the domains. The paper is fairly short and concise, although there are a few issues that should be improved or made clearer.*
*I therefore think minor revisions are necessary, although my questions on the relatively short simulation period may require more effort than the other comments.*

*Abstract*
*The abstract should contain some more on the motivation for including stratospheric chemistry in CAMS. Is forecasting skill part of the motivation?*
*It is stated that this is a first step, which makes me wonder how far away the next step really is.*

The essential motivation for including stratospheric chemistry within C-IFS is to enable the evaluation of ozone and methane oxidation chemistry processes in the stratosphere, to achieve an improved representation of ozone, water vapour and related trace gases within IFS. Combined with the availability of long-lived trace gases such as $N_2O$, evaluation of such an atmospheric composition analysis and forecast system is expected on the long term to lead to improvements to IFS meteorological forecasts as well.
The reviewer is correct that the wording *'first step'* potentially implies to the reader that several major next steps are still needed before operationalization. However, in our conclusions we list the steps that are still foreseen, of which none of them prevents such operationalization. Also an extension of the simulation from 1.5 year to 2.5 year (see also below) showed that the stratospheric trace gas composition remains bounded, with no obvious drift. Even though we acknowledge the system is not yet perfect overall a clear improvement is obtained compared to the current system. Therefore, to

clarify the status of this work, we now judge this as the 'key' step towards operationalization, where essentially the next steps concern the acceptance of ECMWF of the model performance in operational conditions as well as the coding implementation. Depending on the CAMS schedule, final implementation and testing in the operational environment, including a new meteorological cycle, would be required. We now write in our abstract:

"This marks a **key** step towards a chemistry module within IFS that encompasses both tropospheric and stratospheric composition, **and could expand the CAMS analysis and forecast capabilities in the near future.**"

**1. Introduction**
*Page 1, Line 25: "Also analyses and forecasts of stratospheric ozone directly impact the forecast capabilities of surface solar irradiance" is a bit difficult to understand. Suggest changing to e.g. "Also, the amount of stratospheric ozone directly impact the forecast capabilities of surface solar irradiance, making good stratospheric ozone forecast important".*

We thank the reviewer for this suggestion. We replaced this sentence into:
**"Also, the amount of stratospheric ozone directly impacts the forecast capabilities of surface solar irradiance (Qu et al., 2014), stressing the relevance of good stratospheric ozone forecasts."**

*Page 3, Line 5: "We have developed a strategy": Perhaps better to say "We present here a merging …". There is no need to state that there is a strategy, and the work done is the actual merging.*

We agree with the reviewer not to emphasize our strategy, but simply the approach we have taken in our the work. In this section we now write:
"We have developed **an approach for an optimized** merging the CB05 tropospheric chemistry scheme… "

While in the next section we write:
"In this paper we describe **two merging approaches...**"

**2.1 Stratospheric chemistry**
*Page 4, Line 15: Are the surface area densities (SAD) fixed? No size distribution?*

The surface area density (SAD) field for stratospheric aerosol is constructed from a zonal mean aerosol number density field assuming a constant lognormal size distribution with median radius of 0.07 μm and geometric standard deviation of 1.76. The aerosol number density is taken from SAGE II extinction measurements (Hitchman et al 1994). This is different from the 'Daerden et al, ACP 2007' reference as stated in the manuscript. We apologize for this mis-represenation. For ice and NAT PSCs fixed SADs are assumed as reported in the manuscript. We now write:

"The surface area density of stratospheric aerosols uses **an aerosol number density climatology based on SAGE-II observations (Hitchman et al., 1994).**"

*On the fact that you do not do sedimentation, but parameterise it using exponential decay of HNO3 and H2O, is this a good approximation? If you have a situation of very cold temperatures over a long period, the PSCs may sediment out and the SAD will be reduced. Will the parameterisation cause too large denitrification/dehydration? Figure 9 could perhaps indicate this?*

Indeed Figures 6 and 9 in the original manuscript show lower $HNO_3$ and $H_2O$ in BASCOE-CTM and CIFS-TS than the MLS observations over the south pole, indicating a too efficient removal of $HNO_3$ and $H_2O$ through sedimentation as consequence of the simplistic parameterization based on a temperature criterion. We have worked on an improvement of the PSC scheme, see also the response to reviewer #2.

**2.3 Merging procedure …**
*Page 5, Line 24: 40 hPa as tropopause is very low. Does this occur often? To my knowledge, the tropopause pressure is seldom lower than 80hPa. Using 40 hPa may be of small consequence, and may even cause O3 production to be better represented.*

In practice the $O_3$ and CO concentrations always define the interface between tropospheric and stratospheric chemistry, rather than the 40hPa level. This pressure criterion is only introduced to prevent any spurious detection of tropospheric conditions at the top of the atmosphere where CO ($O_3$) increases (decreases), due to $CO_2$ and $O_3$ photolysis. We specify this now more clearly by adding the sentence:

**"With this definition the associated tropopause pressure ranges in practice between approx. 270 and 80 hPa for sub-tropics and tropics, respectively."**

*Page 5, Line 26: "Specificaly" -> "Specifically".*

We changed this, thank you.

*Page 5, Line 34: 10-day decay rate in the stratosphere. What happens to the lost species? Are they assumed to be converted to aerosols?*

The trace gases affected by this decay rate are currently lost, i.e. they do not contribute to the aerosol, CO or NOy components. Clearly, especially a coupling of tropospheric aerosol to the stratospheric ozone chemistry would be an interesting application, as we also mention in our conclusions section, but this is beyond the scope of the current system. We add a sentence for clarification:

**"These losses are currently not accounted for in the stratospheric chemical mechanisms and do not contribute either to the load of stratospheric aerosols."**

*Page 6, Line 11-14: It would be very interesting to see how chemical composition changes when compared to this test. I think some more info should be given on this, as it could explain chemically why you use two domains.*

Also in response to reviewer #2 we now include an evaluation of this model setup where tropospheric and stratospheric chemistry schemes have been fully merged into one single reaction mechanism, which we refer to as C-IFS-Atmos. In addition we now show model profiles near the tropopause for a selection of components in a new Supplementary Material. Even though we see significant differences in the troposphere for short-lived chlorine and bromine-containing trace gases, the differences between C-IFS-Atmos and C-IFS-TS in the stratospheric composition remain small, as confirmed by the extended model evaluation including C-IFS-Atmos. This can be understood since $N_2O$, methyl chloride, methyl bromide and CFC's, which form the largest sources of NOy, HCl and HBr reservoir trace gases in the stratosphere, are marginally affected by these tropospheric reactions. This aspect is also discussed at the start of Sec. 2.3.

Near the tropopause level, at the interface between tropospheric and stratospheric chemistry in C-IFS-TS, some vertical oscillations are visible in C-IFS-TS for $CH_2O$ (Fig. S2), HBr and Br (Fig. S5), i.e. species with relatively short lifetimes which have a different chemistry in the tropospheric and stratospheric modules ($CH_2O$) or no chemistry in the tropospheric module (HBr, Br). The limited difference between C-IFS-TS and C-IFS-Atmos for other short-lived components ($NO_y$, OH, chlorine-containing trace gases, as well as the long-lived trace gases such as $O_3$, CO, $N_2O$, CFC's and $CH_4$) supports the assumption that the chemical split between troposphere and stratosphere is appropriate for key components.

Also the $CH_3Cl$ (Fig. S4) and $CH_3Br$ (Fig. S5) components do show a limited discrepancy between C-IFS-TS and C-IFS-Atmos in the troposphere, associated with the absence of chemical break-down in the tropospheric part in C-IFS-TS. We acknowledge that an extension of the tropospheric reaction mechanism with halogen species also impacts tropospheric ozone and, e.g., the methane lifetime. We found that differences in tropospheric OH, $O_3$ and related components are generally small, while a closer inspection is beyond the scope of this work.

**2.3.1 Merging photolysis rates**
*Page 6, Line 26-27: How is the interpolation/merging done? Some weighting for different layers?*

The photolysis rates are linearly interpolated with pressure between the BASCOE parameterization in the stratosphere and MBA parameterization in the troposphere, at four pressure levels around a fixed tropopause altitude. This altitude is somewhat different (a bit lower towards the troposphere) than the chemical tropopause level adopted for the selection of the solver.

*Page 7, Line 1: The merging only affects 4 layers, where some are in the troposphere, so I am not surprised the over-all impact is small. Perhaps it could be noted that JO3 does not seem to be used in troposphere? Did you check hourly composition at these grid boxes?*

Please note that the photolytic reaction $JO_3 \rightarrow O^1D$ is clearly also active in the troposphere, see also Table 4, even though the $O^1D$ reaction product in the CB05 tropospheric chemistry scheme is only implicitly accounted for (see also response to ref#2). We now present instantaneous model profiles near the tropopause in the Supplementary Material, where we also include results from the C-IFS-Atmos version. We now add in the manuscript:

"Even though such jumps are undesirable, no visible impact on local chemical composition was found, **for any of the trace gases involved in both tropospheric and stratospheric chemistry, see also Figures S1-S3 in the Supplementary Material. This can be explained by the sufficiently small difference in the photolysis rates at the merging altitude of the photolysis and chemistry schemes, combined with the sufficiently long lifetime of the affected trace gases.**"

*2.3.2 Merging tracer transport*
*This is perhaps not really a merging.*

The reviewer is correct. We have changed the subsection title to "**Tracer transport settings**".

*Regarding stratospheric H2O tracer: So you do not use tropospheric H2O (from q) as source of stratospheric H2O? I would expect some boundary condition is needed, at least in the tropics. How is this treated?*

In fact we do use tropospheric $H_2O$, as constrained by q, to serve as boundary condition for the stratosphere. We clarify this better by changing the sentences as follows:

"While a chemical $H_2O$ tracer is defined in the full atmosphere, in the troposphere $H_2O$ mass mixing ratios are constrained by the humidity (q) simulated in the meteorological model in the IFS **and provide a boundary condition for water vapour in the stratosphere**."

*3.1 Observational data ...*
*Page 8, Line 21: "weighted part": What kind of weight?*

The weighted contribution of tropospheric $NO_2$ to the stratospheric $NO_2$ columns are due to the applied air mass factor. We slightly revised this section and now write more explicitly:

"Stratospheric $NO_2$ columns from SCIAMACHY presented here are in fact total columns **derived by dividing retrieved slant columns of $NO_2$ by a stratospheric air mass factor and contains data over the clean Pacific ocean (180°E - 220°E) only (Richter et al., 2005). Although in this region the contribution of the troposphere to total column $NO_2$ is small, stratospheric column $NO_2$ from SCIAMACHY is still somewhat positively biased by a tropospheric contribution. However, stratospheric air mass factors for $NO_2$ are usually large compared to tropospheric ones, so that the uncertainty resulting from this should only have a minor impact on the data analysis presented in this study.**"

*Page 8, Line 23: What is the model output frequency? It would be helpful to specify this and whether you use instant model values.*

In all cases, including the $NO_2$ stratospheric column evaluation, three-hourly instantaneous model values are used. In the case of SCIAMACHY $NO_2$ observations these model values are interpolated in time to 10:00 LT. We now specify this at the end of Sec. 3.1:

**"Three-hourly C-IFS and BASCOE-CTM output has been interpolated in space and time to match with any of these observations."**

**4 Model evaluation**

*Page 9, Line 14-20: The equatorial low bias and high bias at NH mid-lat could indicate too fast transport away from Equator?*

Indeed potential biases in transport would impact on stratospheric composition. This issue is now discussed in more detail in the revised manuscript (see reply to last comment of second reviewer). We believe the stratospheric ozone biases are also associated to biases in NOx, which tend to be positive over the NH mid-latitude, while neutral and/or slightly negative over the tropics from 2009 onwards (despite the corresponding positive bias in $N_2O$). We add the following sentence:

"**The alternating biases in CIFS-TS and CIFS-Atmos are due to corresponding biases in chemically related species such as $NO_x$ and also to transport issues, as discussed in more detail below.**"

*Figure 2: The simulated time period is not long enough to make clear whether the bias will build up. This should be further explained.*

The reviewer is correct that a 1.5 year simulation is relatively short to assess the evolution of stratospheric composition, and the corresponding development of biases. For that, multi-year simulations are required. We have extended all model runs with one extra year (2010) and expanded the corresponding evaluations. This revealed that the biases in $O_3$ remained essentially the same in 2010 compared to 2009. We now include figures showing time series of $O_3$ partial columns and total columns, as well as $NO_2$ total columns over the full period. However, please note that the purpose of this system is to provide accurate analyses and short-term forecasts of atmospheric composition, rather than good multiyear simulations, as also acknowledged by the reviewer.

*Ozone hole: Do you overestimate PSCs (as commented earlier) and hence halogen activation?*

We revised our PSC-parameterization (see also Ref#2), which indeed led to an improvement in halogen activation. We now expand our assessment of $O_3$, $HNO_3$ and ClO evolution during 2009 Austral spring. In the revised setup the modeled ClO follows very well the observations, although its decrease is about two weeks earlier than observed, as now documented in a new Figure (Fig. 6 in the revised manuscript) Still the $O_3$ appears under-estimated to a larger amount than BASCOE-CTM, suggesting that transport-related elements also contribute to biases in $O_3$.

*Page 9, Line 20-32: Is the comparison done by extracting the same profiles (location and time) as in sondes? Should be specified.*

For our methodology for comparing the model ozone with the sondes we use three-hourly model output which has been interpolated in time and space to match with the observations. We now specify this in the manuscript, as discussed above.

*The sonde comparison should include some variability in sondes (e.g. as standard deviation), and possibly also from the model (Fig. 3-5). (In Figure 3-4 this could be placed in either upper or lower row.)*

We now present the standard deviation of the variability in the $O_3$ profiles, both for the observations and the collocated model results. Also, in response to reviewer #2 Figures 3 and 4 now present the $O_3$ evaluation in terms of ppmv rather than mPa. We have adapted the figure legends accordingly.

*Figure 6: How is OH in IFS vs BASCOE? Do you have less HNO3 in stratosphere because of hotter photochemistry? Some thoughts/explanation should be given for the higher low-SZA NO2 in IFS than in BASCOE. NO2+NO3 <-> N2O5 at night? Photochemistry?*

The OH between C-IFS-TS and BASCOE-CTM is essentially identical, as largely governed by $O^1D$ reaction with $H_2O$, and in turn on $O_3$ abundance and photolysis. This indeed does not give rise to the differences in $HNO_3$ between C-IFS-TS and BASCOE-CTM, which occur mainly at an altitude of approx. 10-30 hPa. Considering the equal photochemistry in the stratosphere the reduced $HNO_3$ in C-IFS-TS compared to BASCOE-CTM the discrepancy points at transport-related features, as also discussed with Figure 9. Note that the (reasonable) daytime $NO_2$ maximum, which is found at an altitude of ~10 hPa, is not directly related to this aspect, nor to the anomalously higher $NO_2$ during night-time (i.e. at high SZA) in C-IFS-TS compared to BASCOE-CTM and observations. This positive offset is largely occurring in the 5-1 hPa altitude range, but remains limited to biases in daytime NO and nighttime $NO_2$ only, while other trace gases in this altitude range ($N_2O$, $NO_3$, $N_2O_5$, $HNO_3$) do not explain this bias. The following sentence has been added at the end of the paragraph:

**Even though a clear improvement compared to run C-IFS-T is found, further investigation is necessary to diagnose the origins of the biases in night-time $NO_2$ above 10 hPa and in $HNO_3$ between 10 and 70 hPa.**

*Page 10, Line 3-4 (Figure 7): NH higher $NO_2$: Could a possible reason be that SCIAMACHY assumes that a too high fraction of the column is located in the troposphere?*

The fact that model columns are generally larger than retrieved ones cannot be explained by the contribution of the troposphere to stratospheric columns from SCIAMACHY alone, as stratospheric columns from SCIAMACHY are positively biased by a tropospheric contribution to a minor extent only (see our earlier response above). Moreover, the overestimation compared to SCIAMACHY retrievals cannot be explained by error estimates for SCIAMACHY stratospheric $NO_2$ columns given in the manuscript (relative uncertainties of roughly 5-10% and additional absolute uncertainty of $1 \times 10^{14}$ molec $cm^{-2}$) alone, which only account for about $0.3 \times 10^{15}$ molec $cm^{-2}$ of the positive bias compared to SCIAMACHY. Hence the slight positive bias suggests a C-IFS model issue, as is also confirmed with the $NO_2$ evaluation against MIPAS observations in Figure 6, where the model also shows larger values than the observations over the NH, around the 8hPa altitude.

*Page 10, Line 19: What about too fast horizontal transport? Is CH4 fixed at surface?*

We constrain $CH_4$ at the surface using a monthly and latitudinally varying climatology, which is somewhat different than the null flux approach adopted in BASCOE-CTM. Nevertheless, the agreement with BASCOE-CTM (and observations) at ~ 100hPa does not suggest issues with the tropospheric $CH_4$ concentrations that could explain the discrepancies seen in the stratosphere. Indeed not only vertical transport, but also horizontal transport and mixing could be causes for differences between C-IFS and BASCOE-CTM in the 100-10 hPa altitude range. Thanks to the last comment by second reviewer, the discussion of the transport issue indicated by fig.9 has been completely re-written in order to avoid any over-interpretation:

**"Fig. 9 shows an evaluation of $N_2O$ and $CH_4$ profiles during September 2009 against observations by ACE-FTS. Owing to their long lifetimes these trace gases are good markers for the model ability to describe transport processes - i.e. not only the Brewer-Dobson circulation but also isentropic mixing, mixing barriers, descent in the polar vortex, and stratosphere-troposphere exchange (Shepherd, 2007). Moreover, $N_2O$ is the main source of reactive nitrogen in the stratosphere while $CH_4$ is one of the main precursors for stratospheric water vapour. The figure suggests reasonable profile shapes for both $CH_4$ and $N_2O$ in the upper stratosphere (10 hPa and higher) where their abundance is more strongly influenced by chemical loss but at lower altitudes (100-10 hPa) C-IFS-TS and C-IFS-Atmos show larger discrepancies to the observations than the BASCOE-CTM run, with weaker vertical gradients in the tropics and SH-mid latitudes and a sharper gradient in the extra-tropical Northern Hemisphere.**

**This discrepancy cannot be due to different wind fields because the BASCOE CTM experiment is driven by three-hourly output of the C-IFS-T experiment. We attribute it instead to the different numerical schemes for advection and/or to differences in the representation of sub-grid transport processes in the GCM and in the CTM. Convection and diffusion are indeed explicitly modelled in C-IFS but neglected in BASCOE CTM, which relies on the implicit diffusion properties of its flux-form advection scheme to represent sub-grid mixing (Lin and Rood, 1996; Jablonowski and Williamson, 2011).. Since lower stratospheric ozone is strongly determined by both chemistry and transport, the transport issue indicated by Fig. 9 could also contribute directly to the ozone biases seen below 10 hPa in Figures 3 and 4."**

*5 Conclusions*
*Page 11, Line 3: I generally do not think 1.5 years is enough for evaluating the chemistry and chemistry. A possible drift in the O3 column (Fig. 2) should be investigated. If chemistry is adjusted in an assimilation system, a drift will probably not be very prominent or important, but used as a CTM 1.5 years is short.*

We sympathize with the comment of the reviewer that a 1.5 year simulation is too short to assess and quantify potential drifts. Therefore we have now extended the runs and corresponding evaluation with an additional year (2010). This indeed shows that biases in $O_3$ partial and total columns for 2010 are

essentially similar to 2009, and also the Antarctic $O_3$ hole period was modeled with similar skills. We extended the discussion in the full manuscript on this additional year.

*Page 11, Line 9-11: "a larger error" -> "larger errors", "was" -> "were", and fix rest of sentence.*

We changed this, thank you for the suggestion.

*Page 11, Line 12: Why is it necessary to do this first step? It seems another step is expected.*

See also our comments above. We now modify this as follows:

"This benchmark model evaluation of C-IFS-TS marks a **key** step towards merging tropospheric and stratospheric chemistry within IFS, aiming at **a possible configuration for** daily operational forecasts of lower and middle atmospheric composition **in the near future**."

*Page 11, Line 18: What use would assimilation of long-lived species have in the IFS?*

Assimilation of long-lived species ensures the provision of observationally constrained trace gas fields which will intrinsically contribute to an improved quality of ozone chemistry specifically, and atmospheric composition forecasts in general, which is one of the essential purposes of the system.

*Page 12, Line 6: While monitoring capabilities may be important and interesting in the stratosphere, and also provide global products of species which are not globally observed, it would be interesting to hear the implications for forecasting.*
*Is it not only O3 that is important for radiation calculations? Better stratospheric O3 could improve stratospheric temperatures?*

Indeed better stratospheric $O_3$ is one of the key target products which are expected to lead to improved stratospheric temperature fields, as shown e.g. by de Grandpré et al. (2009). Also stratospheric water vapour which is also highly relevant to radiation, can potentially benefit from an improved representation of stratospheric chemistry. Note that improvement in description of these tracer fields can either be obtained through revised climatologies, as derived from stratospheric composition analyses, as well as through revised parameterization of prognostic variables.

We explicitly hint on these aspects in the introduction of our manuscript, as one of the motivations to develop this system. As for the conclusion, we believe the general statement '(…) stratospheric chemistry (…) may also contribute to advances in meteorological forecasting of the ECMWF IFS model in the future' is appropriate, as at this stage it is too early to specify such applications.

***Appendix***
*Table A1 should be sorted on names.*

The reviewer is correct that the ordering was not optimal. We have re-ordered the list of trace gases, first grouping trace gases active in the various regions (glb, trop, strat), and next sorting them more strictly on functional groups (e.g. grouping the hydrocarbons, the chlorine-containing trace gases, etc.)

**References**
*DOIs are missing for some references; please update.*

We have updated the list of references, including the missing DOIs. The additional references in the revised manuscript are listed after the reply to the second reviewer.

---

## Author Comment (AC2) · 8 Jul 2016

Request:

Please add a version number or unique identifier either for each of the models you combined or for your newly created model in the title upon your revised submission to GMD.

Response:

We thank Dr Kerkeweg for referring to the requirements of papers published in GMD. In fact also the editor of this manuscript, Dr Grewe, had pointed us to these requirements, where we responded that version labeling was not yet introduced, in line with

arguments given by Flemming et al. (2015): The cycle number of the IFS reflect the development of the NWP code, but are not yet linked to the development of the chemistry modules. The current chemistry scheme is used for several IFS cycles. As part of the Copernicus Atmosphere Monitoring Service a proper version naming convention will be introduced. In case our response is not acceptable for acceptance in GMD we will now introduce such a version number to this new model implementation.

––––––––––––––––––––––––––

---

## Author Comment (AC3) · 8 Jul 2016

**Response to the reviewer comments on the manuscript**

**C-IFS-CB05-BASCOE: Stratospheric Chemistry in the Integrated Forecasting System of ECMWF**

By V. Huijnen et al.

First, we would like to thank the reviewers for their critical, but useful comments. In view of their valuable suggestions in our revised manuscript we have:

1. included an additional model configuration containing full (tropospheric *and* stratospheric) chemistry within the whole atmosphere
2. revised our PSC-parameterization
3. extended our model evaluation with one additional year
4. revised some of our evaluations

The reviewer's comments are given in italic, and our responses in regular font. Textual modifications to the manuscript are highlighted in bold. Figure numbers refer to the revised manuscript.

**Response to anonymous Referee #2**

*The purpose of the paper is to describe and benchmark a new version of the IFS model. This version has separate chemistry modules (mechanisms) for the troposphere and stratosphere, where the decision of which module to call is determined by the altitude of the grid box with respect to the tropopause. The stratospheric chemical mechanism comes from an assimilation system (BASCOE). The previous version of the IFS had*
*tropospheric chemistry plus linearized stratospheric O3. The paper concludes that a new simulation that uses both chemical mechanisms (called CIFS-TS) has good stratospheric O3, NO2 and other reactive trace gases compared to satellite data sets.*

*A goal of the paper is to demonstrate that their method of using the tropospheric mechanism/solver for tropospheric grid boxes and the stratospheric solver for stratospheric grid boxes is a computationally efficient way to calculate the full chemistry of the atmosphere. The biggest problem with this paper is that they did not actually test this. To demonstrate their method, they need to have run a simulation where tropospheric and*
*stratospheric reactions were solved TOGETHER and NOT split into 2 mechanisms.*
*Those results could then be compared with their 'split' method. Ideally, this would show that their method was faster (how much faster?) yet produced essentially the same results. I recommend they do this and then rewrite this manuscript. This experiment would not only satisfy the stated goal of the paper but it would also eliminate the confusion in the comparisons (see below) regarding transport/advection differences between BASCOE-CTM and the CIFS-TS.*

We thank the reviewer for his/her valuable comments on our manuscript. Indeed a system with tropospheric and stratospheric chemistry resolved throughout the atmosphere (in the remainder referred to as 'C-IFS-Atmos'), as opposed to the reported more efficient approach in C-IFS-TS, had been already developed and briefly mentioned in the manuscript but its evaluation was missing. The main reason for not presenting this was that the stratospheric chemistry was treated very similar and hence small differences in model results between C-IFS-Atmos and C-IFS-TS could be expected in the

stratosphere. Discrepancies to the observations mainly raise from common stratospheric chemistry model assumptions (e.g. PSC treatment, photolysis) and differences in transport treatment between C-IFS and BASCOE-CTM. Larger differences can be only be expected when approaching the tropopause. We would like to point out that the original manuscript does mention that C-IFS-Atmos is 50% more expensive than C-IFS-TS (at the end of Sec. 2.3), essentially due to the larger chemical mechanism throughout the atmosphere that needs to be solved.

In response to the reviewer we acknowledge that an explicit evaluation of the differences between C-IFS-TS and C-IFS-Atmos does clarify our goal, which indeed also aims at presenting our methodology with separate tropospheric an stratospheric chemistry in C-IFS-TS. We therefore now include explicitly C-IFS-Atmos in our model evaluation and show that the differences with the more efficient approach in C-IFS-TS are as small as expected.

This expansion of the model evaluation also increases the usefulness of the comparison with BASCOE-CTM since it is now possible to compare two models with the same stratospheric chemistry but different transport schemes (BASCOE-CTM versus C-IFS-TS) and two models with the same transport but (slightly) different chemical schemes in the stratosphere (C-IFS-TS versus C-IFS-Atmos). A recent study by de Grandpré *et al.* (2016) is now cited to illustrate the type of issues raised in the stratosphere by the semi-lagrangian advection scheme.

*I don't agree with the statement in the abstract that the new model configuration shows good performances of stratospheric O3, NO2, and other tracers. The figures chosen to demonstrate good representation of various stratospheric constituents in the CIFS-TS model generally show fair to poor agreement with observations. Stratospheric O3, for example, often looks worse (or at least no better) that it did in the CB05 (trop only) or BASCOE (strat only) versions. This new model does not appear to be an improvement over previous model versions.*

*The comparisons between CIFS-TS and BASCOE-CTM are confusing. When stratospheric species such as NO2, HNO3, and O3 are compared, the results are different. I thought the primary goal of the paper was to compare the chemical mechanisms, but since the results are rather different, there must be transport (or meteorological field) differences too. This is alluded to on page 7, lines 26-28. The transport/advection needs to be the same in the two simulations in order to compare the chemical mechanisms.*

*It should be made clearer in the text what the differences are between CIFS-TS and BASCOE-CTM. My overall recommendation is to test a combined (strat+trop) solver and compare the results to trop only, strat only, and the 'split' method presented here. The results will provide a good benchmark and will be easier to interpret if all experiments are performed with the same transport code and meteorological fields.*

The reviewer appears confused by the selection of model setups chosen in our manuscript. In essence, the C-IFS is a Global Circulation Model (GCM) designed for meteorological analyses and forecasts where a module for chemistry has been included to extend its abilities in terms of atmospheric composition. On the other hand, the BASCOE system is a dedicated data-assimilation system for stratospheric composition, based on a Chemistry Transport Model (CTM) environment, i.e. a completely independent system to C-IFS.

The impact of various chemical mechanisms was evaluated though comparison of C-IFS-T (with linear ozone treatment in the stratosphere) and C-IFS-TS (which uses the identical chemical parameterization in the stratosphere as BASCOE-CTM).

The BASCOE-CTM is driven by meteorological fields from the C-IFS run, but still uses a different numerical scheme for the advection, and is running on a different grid as compared to the C-IFS runs. Note that in this setup of the BASCOE system the chemical data-assimilation is switched off, hence purely reflecting the forward model capabilities. Hence comparison between BASCOE-CTM and C-IFS-TS is a clean method to evaluate differences due to the representation of transport with identical meteorological fields, and not suited for the evaluation of differences in the chemical treatment since there are no such differences.

The additional model run, C-IFS-Atmos, where tropospheric chemistry is extended throughout the stratosphere, and vice versa, is now included to assess the impact of assumptions of the reduced chemistry in C-IFS-TS.

In response to the reviewer's concerns we now extend the table describing the model versions (see also the comment below), and extend the description between differences of the various setups. Notably in the introduction we now write:

**"The CB05 tropospheric scheme has been combined with the stratospheric scheme from BASCOE-CTM to form a single chemistry mechanism that encompasses tropospheric and stratospheric chemistry throughout the atmosphere, here referred to as C-IFS-Atmos. However, this approach appears computationally expensive, due to the extended chemical mechanism. Therefore …."**

And also:

**"In this optimized approach we developed** a flexible setup where -within a single framework- either the tropospheric or stratospheric chemistry modules are addressed**, referred to as C-IFS-TS. In this approach the parameterizations for the chemistry, including the respective chemistry mechanisms as optimized for troposphere and stratosphere separately, are retained.**
In this paper we describe our **two merging approaches** and provide benchmark evaluations of the **C-IFS-Atmos and** C-IFS-TS systems with focus on the stratospheric composition**. The ancestor BASCOE-CTM is also included in the comparison through a forward model run (without chemical data assimilation) in order to provide insight in the differences caused by the treatment of transport between C-IFS and BASCOE."**

The model evaluation has been extended to include results obtained with C-IFS-Atmos, as well as an evaluation of the stratospheric composition (including $O_3$, $HNO_3$ and $NO_2$) in C-IFS-T, to explicitly identify the impact of the newly implemented stratospheric chemistry within the C-IFS framework.

***Other points***
*It would be helpful to add a table that lists the specifications of each of the models used and notes how dynamical fields are obtained (e.g., forecast, assimilation, . . . ?), chemical mechanism, resolution, etc. For example, BASCOE is an assimilation system, but it's only the BASCOE stratospheric chemical mechanism that used here, right? And BASCOE-CTM means the assimilated (renanalysis) fields have been saved and then*
*are being used in an offline chemistry transport model? Presumably it is the same offline model that the C-IFS forecast fields are used in? If what I am asking does not make sense, please take this as an indication that I am confused by the descriptions of the models.*

In response to the reviewer's concerns in Section 2.3 we expanded Table 2 which lists the specifics of the various model systems, as also given below. Further, we want to make clear that the C-IFS experiments have been run in 'nudged meteo' mode, by relaxation of the meteorology towards ERA-Interim, as we also write in Section 3. The BASCOE-CTM run is driven by the identical meteorology from the C-IFS experiment (and in turn from ERA-Interim), but applies its own advection algorithm which is clearly different from the one used in IFS. In Section 3 we also make more clear what are the differences between C-IFS and BASCOE-CTM. As discussed above, the BASCOE-CTM results are included as a reference of what can optimally be achieved with C-IFS-TS and C-IFS-Atmos in the stratosphere, using only simulations nudged with specified dynamics and unconstrained composition. Specifically we now write:

"Meteorology **in the C-IFS runs** is relaxed towards ERA-Interim (…) The performance of the C-IFS runs has further been compared against the BASCOE-CTM (without **chemical** data assimilation), using the same chemical mechanism and parameterizations **for photolysis and heterogeneous chemistry** as implemented in the C-IFS-TS. **This serves as a model reference for the C-IFS implementation of stratospheric chemistry. While C-IFS evaluates tracer transport on a reduced Gaussian grid, the** BASCOE-CTM **uses a regular latitude-longitude grid. It** is run **here** with a resolution of 1.**125°** lon / lat similar to the resolution **chosen for** C-IFS, and on the same vertical grid of 60 levels**. The BASCOE-CTM is driven by** temperature, pressure and wind fields simulated by the C-IFS runs. **However, while BASCOE adopts a flux-form advection scheme (Lin and Rood, 1996) the IFS uses the Semi-Lagrangian scheme for advection, accounts for vertical diffusion and includes a parameterization for convection (ECMWF, 2015).** Using essentially the same dynamical fields together with an identical implementation of the chemistry code should allow to identify differences **due to the different** transport schemes between C-IFS and the BASCOE-CTM. Common chemical biases between both systems also point at issues in the chemical parameterization**s such as reaction mechanism, photolysis, heterogeneous chemistry and sedimentation.**"

**Table 2.** Number of trace gases, the chemistry scheme in troposphere and stratosphere, and corresponding number of reactions (gas-phase / heterogeneous and photolytic), as well as specification of the circulation model and computational expenses of a one-month run on T255L60 in terms of system billing units (SBU) for various C-IFS model versions. For completeness also the BASCOE-CTM system is indicated.

| | C-IFS-T | C-IFS-S | C-IFS-Atmos | C-IFS-TS | BASCOE-CTM |
|---|---|---|---|---|---|
| **No. trace gases** | 55 | 59 | 99 | 99 | 59 |
| **Chemistry scheme in troposphere** | CB05 | BASCOE (P<400hPa) | CB05+BASCOE | CB05 | BASCOE (P<400hPa) |
| **Chemistry scheme in stratosphere** | CB05/ Cariolle | BASCOE | CB05+BASCOE | BASCOE | BASCOE |
| **No. reactions (gas / het / photo)** | 93/3/18 | 142/9/52 | 211/11/60 | 93/3/18 *or* 142/9/52 | 142/9/52 |
| **Circulation model** | GCM | GCM | GCM | GCM | CTM |
| **SBU** | 2075 | 2500 | 4563 | 3076 | - [a] |

[a]**BASCOE does not run on the ECMWF supercomputing facility and hence cannot be compared directly to C-IFS in terms of computational resources.**

*Regarding 'tracer species' or similar expression found in many places, 'tracer' means a species that is unreactive and can be used to trace something, like transport. I think you mean 'trace gas' rather than tracer because that can be used in a general way to talk about any type of constituent in the model. Please search on 'tracer' in the document to identify where you mean trace gas or constituent.*

We thank the reviewer for this comment, and changed the wording accordingly throughout the document.

*p. 3, l.24. Are you saying the chemistry in the modules is parameterized? Or are you referring to the chemical mechanisms when you say 'chemical parameterization'? A parameterization for chemistry is not the same thing as a chemical mechanism. Sometimes 'chemical schemes' is used, which is fine for referring to the mechanism. This confusion occurs throughout the paper. Please check each occurrence of 'parameterization' to verify the right words were chosen.*

In this occasion the phrase 'chemistry parameterization' referred to all chemical conversion processes that require a parameterization, including aqueous phase and heterogeneous reactions as well as photolysis and parameterizations for sedimentation. Indeed this refers to more than just the definition of the chemical mechanism. To accommodate the concerns of the reviewer we had a critical look at our terminology for 'parameterization' in the complete manuscript, and changed it where appropriate (see also below). In this instance mentioned by the reviewer in Sec. 2.0 (p.3, l.24) we only wish to guide the

reader forward to the specific sections on stratospheric/tropospheric chemistry, but to prevent potential confusion we now write "tropospheric (CB05-**based**) chemistry parameterizations".

*p. 4, l. 15. The threshold temperature for NAT formation is pressure dependent. The manuscript indicates that 194 K was chosen as the threshold regardless of pressure. That would not be the correct way to calculate it.*

While the BASCOE CTM was used some time ago for detailed studies of the processes leading to polar ozone depletion (Daerden et al., ACP, 2007), the corresponding microphysical module was removed (due to huge computational costs) and replaced by this very crude parameterization. Indeed the BASCOE CTM is now designed as a generic model which (until now) needs only to be good enough to allow the successful assimilation of satellite observations of stratospheric composition. Yet both reviewers indicated a simple improvement which could be implemented quickly enough fot this revision of the manuscript.

Hence we have revised the PSC-parameterization, which is no longer purely temperature-dependent. We now remove $H_2O$ and $HNO_3$ where their respective partial pressures exceed the equilibrium values, according to Murphy and Koop (2005), and Hanson and Mauersberger (1988). The time scale for irreversible removal of $HNO_3$ has been revised from 100 days in the original setup to 20 days, in accordance with the smaller regional and temporal extent where NAT particles are assumed to exist. This led to significant improvements in the $H_2O$ and $HNO_3$ bias in the region where PSC formation is possible, and accordingly to a slight improvement in $O_3$ profile shapes in terms of a reduced positive bias at 100 hPa and reduced negative bias at 20hPa during August-September over the Neumayer and Syowa stations (see also below). Nevertheless, the $HNO_3$ timeseries for the BASCOE-CTM, CIFS-TS and C-IFS-Atmos models suggest that denitrification proceeds more slowly and ends one month later than observed by Aura MLS observations, which may be attributed to our crude modelling approach for the formation and sedimentation impact of NAT PSC.. We have modified the respective section as follows:

"Ice PSCs are presumed to exist at any grid point in the winter/spring polar regions where **water vapour partial pressure exceeds the vapour pressure of water ice (Murphy and Koop, 2005).** Nitric Acid Tri-hydrate (NAT) PSCs are assumed **when the nitric acid ($HNO_3$) partial pressure exceeds the vapour pressure of condensed $HNO_3$ at the surface of NAT PSC particles (Hanson and Mauersberger, 1988)."**

*p. 5, l. 31. I don't understand what is meant by O1D and O3P being described implicitly, as opposed to being treated explicitly.*

Within the troposphere the $O^1D$ is produced from $O_3$ photolysis and assumed to react instantaneously, with only reaction products $H_2O$ and again $O_3$. As the $O^1D$ (and O) lifetime is much shorter than the integration time, while only reactions with N2 and O2 are assumed in the troposphere, the $O^1D$ concentration can be considered in equilibrium over the integration time and hence does not need to be treated explicitly. The same argumentation holds for $O^3P$, produced from $O_2$ photolysis in upper troposphere, and assumed to only react with $O_3$ to form $O_2$, and with $O_2$ to form $O_3$. This is different for

the stratosphere, where O1D and O3P are involved in many more reactions. To clarify in the manuscript we reformulate this as follows:

"It is worth mentioning that the constituents $O^1D$ and $O^3P$, produced from $O_3$ and $O_2$ photolysis, **are not explicitly computed in the troposphere, as $O^1D$ and $O^3P$ are assumed to react with $O_2$, $O_3$ and $N_2$ only. This is different for the stratosphere, where $O^1D$ and $O^3P$ are involved in many reactions.**"

*p. 6, l. 24. 'solar radiation reaches the stratosphere earlier than the surface. . .' as written this sounds like it is referring to delay caused by the speed of light! I doubt this was intended; it needs better wording.*

The reviewer is clearly technically correct. We changed the formulation to a more compact formulation, leaving out the suggestion of a different timing:

"Also the presence of sunlight at solar zenith angles (SZA) larger than 90° at high altitudes needs to be accounted for in the stratosphere **due to the Earth's curvature, but may be neglected** in the troposphere. **This plays a role in the timing of springtime ozone depletion in the polar lower stratosphere**. "

*p. 7, Section 2.3.1. For JO3, the lack of a 'jump' in O3 may be because photolysis is unimportant (slow) near 100 hPa, so O3 is probably long-lived relative to the photochemical lifetime. JNO2 is much larger so I'm not sure why there isn't a jump – can you explain this? It would be useful if you showed the simulated O3 and NO2 profiles in Fig. 1 to demonstrate the lack of a jump. What is the meaning of 'JO3_TB' in the title of one plot? No similar title for the other plot.*

The reviewer is correct in that the presence or absence of jumps associated to the change in the reaction mechanism depends on the lifetime of the species, in combination with the magnitude of the change in the dominating reaction (or photolysis) rate with the different chemical mechanism. For $O_3$ the photolysis is a dominating loss term in this altitude range, but still the reaction rate is sufficiently low (i.e. the $O_3$ lifetime sufficiently long) such that jumps in the photolysis rate do not lead to jumps in $O_3$ concentrations. For $NO_2$ the photolysis rate is much larger, and resulting in a short (less than 1 hour) $NO_2$ lifetime. Jumps in photolysis rate potentially result in jumps in NO and $NO_2$ concentrations. Nevertheless, the jump is sufficiently small (for J-$NO_2$ we verified that the difference in photolysis rates around the tropopause is generally below 5%), such that the $NO_2$ concentrations do not show a significant jump. We now provide figures in the supplementary material where we present instantaneous profiles of a range of trace gases at the tropopause interface. We extended the discussion on this aspect with the following sentences:

"Even though such jumps are undesirable, no visible impact on local chemical composition was found, **for any of the trace gases involved in both tropospheric and stratospheric chemistry, see also Figures S1-S3 in the Supplementary Material. This can be explained by the sufficiently small difference in the photolysis rates at the merging altitude of the photolysis and chemistry schemes, combined with the sufficiently long lifetime of the affected trace gases.** "

*Section 2.3.2, l. 8. It's unclear whether you're saying NO, NO2, and have the mass fixed applied or whether they are the few species where the mass fixed isn't applied. How badly is H2O not conserved in the stratosphere? This will conceivably cause problems for stratospheric chemistry. It would be useful to see a 1-year time series of the H2O mass above 100 hPa.*

As explained in the manuscript the reason for switching off the mass fixer for the stratospheric $H_2O$ tracer is because otherwise mass conservation errors originating from the troposphere lead to spurious redistribution of $H_2O$ mass towards the stratosphere. Therefore, in fact due to switching off the mass fixer, the $H_2O$ mass in the stratosphere remains very stable. We illustrate this by Figure R1 (below), which shows indeed absence of any trend in stratospheric $H_2O$ columns over the years, indicating that $H_2O$ mass conservation is sufficiently well ensured in the stratosphere. This figure also shows that $H_2O$ total columns are essentially identical in C-IFS-Atmos and C-IFS-TS.

[Figure]

**Figure R1.** Evolution of global, daily mean $H_2O$ partial columns (left: 0.1-80 hPa, right: 0.1 – 100 hPa) for the runs C-IFS-Atmos (blue) and C-IFS-TS (orange) for January 2009 to December 2010. C-IFS-TS is on top of C-IFS-Atmos.

In the manuscript we now write:
**"The global advection errors in $H_2O$ that essentially originate from the tropospheric part because by far most $H_2O$ mass is located in the troposphere and the spatial gradients are much more pronounced. This should not affect the stratospheric $H_2O$ mass budget, therefore the global mass fixer for the stratospheric $H_2O$ tracer has been switched off. This prevents spurious trends in stratospheric $H_2O$ columns over the years (not shown), indicating that $H_2O$ mass conservation is well ensured in the stratosphere."**

*p. 7, last 3 lines. This sentence says you are looking to identify differences in transport schemes. This confuses the issue of evaluating the chemical mechanisms (and their implementation). This evaluation should be performed using the same dynamical fields with the same model. If the advection schemes are also different, then we cannot actually test the impact of chemical mechanisms alone. And does 'parameterization' in line 28 refer to the different chemical mechanisms?*

For a discussion on the selection of the model setups evaluated in our manuscript we refer to our response to the reviewer's first general comment. We now extend the evaluation with results from run C-IFS-T, to explicitly identify the impact of the newly implemented stratospheric chemistry within the C-IFS framework. Indeed, the BASCOE-CTM run uses identical chemistry to C-IFS-TS and is not introduced to assess the chemical mechanisms, but rather differences due to the transport scheme while using the

same dynamical fields. Here, the 'parameterizations' refer to the reaction mechanism, photolysis, heterogeneous chemistry and sedimentation, as we now explicitly write.

*p. 8, l. 27, 'first Science Satellite'?*

Indeed this is the meaning of the abbreviation 'SCISAT-1'.

*p. 8, l. 30-31. Suggest to change to '. . .between 6-30 km agree to within 15% of independent . . ." For all the figures that are line plots (starting with Figure 2), the blue and black lines are hard to distinguish. Please do something with the line thickness and colors to improve readability.*

We changed this according to the reviewer's suggestions, thank you. We have improved color-coding and general figure quality, which unfortunately also had seen some degradation in the stage of pdf-generation from the word-document.

*Section 4, Model Evaluation p. 9, lines 14-19. This paragraph would benefit by a general statement of the purpose of this comparison. It appears the purpose is to show that the TS mechanism looks more like the observed total column O3 than does the trop-only code (with linearized strat O3). One would expect the TS O3 to be better than the linearized O3 of CB05, but there should also be a comparison with the stratonly code. Comparing with the O3 results in Fig. 6, I think the strat chem O3 columns would be lower than the TS mechanism. I guess they aren't the same because the BASCOE-CTM has different transport. Again, not having the same transport in all the simulations really interferes with a useful comparison.*

We now replace this figure with an evaluation of the partial columns (10-100hPa) against Aura MLS observations, to emphasize the performance in the stratosphere. We now also include results from C-IFS-Atmos and C-IFS-T, as well as from BASCOE-CTM to assess the impact of different chemistry approaches, and different transport scheme. The new evaluation shows more clearly the benefits and limitations of the new approach in C-IFS-TS, as compared to C-IFS-T (with linearized O3), as well as differences with BASCOE-CTM (which contains stratospheric chemistry only and uses the same dynamical fields as C-IFS but with a different transport algorithm). We have moved the assessment of $O_3$ total columns against the Multi-Sensor Reanalysis to the Supplementary Material . This does not include results from BASCOE-CTM, considering it's missing tropospheric contribution. The manuscript text has been revised accordingly.
Furthermore, in Sec. 3 we now include a few general statements to clarify the purpose of the various model evaluations:

**"Intercomparison of the runs C-IFS-TS and C-IFS-Atmos aims to provide a justification of our approach to split the chemistry into two regions, while intercomparison of C-IFS-TS with C-IFS-T can be used to identify the changes to stratospheric composition modelling between full stratospheric chemistry and the baseline approach with the linear ozone scheme."**

*p. 9, discussion of Figs. 3-4. I do not agree that there are meaningful, reduced biases in the TS version. The linearized O3 chemistry of the trop mechanism gives different results from the TS version, but not*

*really worse. These figures show that TS is not an improvement over trop-only. I think the use of mPa for the O3 bias (lower panels) is misleading and probably minimizes the appearance of the disagreement in the middle stratosphere.*

By evaluating $O_3$ profiles in terms of partial pressure biases in the original manuscript we intended to focus on the contribution of each pressure region to the $O_3$ TC, with larger weights in the lower stratosphere. This is now assessed in detail in the revised Figure 2 that presents the evolution of the $O_3$ partial columns (10-100hPa). Hence in accordance with the reviewer request we now present results of $O_3$ profiles in units ppmv, indeed giving more focus to the altitude with maximum $O_3$ concentration, at around 10 hPa. Also we now average over all profiles in 2009 and 2010, to improve the statistics, and include results from C-IFS-Atmos. We agree with the reviewer that we have been too positive when describing the C-IFS-TS results as compared to C-IFS-T (with linear chemistry). We have rewritten this section to provide a more balanced discussion.

*p. 9, discussion of Fig. 5. I cannot tell the difference between obs and CIFS-T lines in the figure. There is no line color/style for the observations in each panel's legend. The TS O3 agrees with one of the black lines (obs or CIFS-T??) near and below 100 hPa – sometimes – but the TS O3 consistently has poor agreement above 50 hPa. Why? Since the TS (red) line often does not agree with either black line – I see no basis for claiming good agreement. Additionally, Syowa is often near the vortex and has large daily variability. Were the simulated profiles used in this figure calculated from the same days of the month as the Syowa data?*

Figure 5 in the manuscript has been regenerated based on the revised model simulations. Color-coding has been updated, and error-bars denoting the model and observation variability are now included. Note that all comparisons with observations, including fig. 5, use three-hourly model output which has been collocated in time and space to the observations. This is now explained in Section 3. Please also note that in our section describing Figure 5 we do not claim general good agreement, as the reviewer suggests, but rather point at regions and months where C-IFS-TS performs well, and others where it shows biases compared to observations.
The revised simulations have seen some improvement in terms of vertical profile shape during ozone hole conditions, see also Figure R2, below, for an assessment of the differences to the C-IFS simulations presented in the GMDD manuscript. This is due to the improved PSC parameterization, especially above 50hPa in August and September where PSCs were allowed to exist in the C-IFS-TS run for the original GMDD version. The remaining discrepancies could still be caused by the limitations of the revised PSC parameterization. We now write:

"For the 2009 Antarctic ozone hole season (Fig. 5) the C-IFS-TS **and C-IFS-Atmos** shows a positive bias at ~100 hPa for August and September, **and negative bias at higher altitudes (50-10 hPa), where C-IFS-T shows a positive bias**."

Additionally we now provide a closer analysis of the performance during polar ozone depletion, by presenting time series of $HNO_3$, ClO and $O_3$ during the 2009 ozone depletion over Antarctica (the new Figure 6). This new figure clearly shows the abilities and limitations of the different versions of C-IFS to describe this event. Specifically we now show that denitrification, which is clearly not modelled in C-IFS-T, starts at the correct time in the models with stratospheric chemistry, although it appears to last about one month too long as compared to the observations. We note that in the original manuscript, where this parameterization depended only on T, the denitrification started one month too late.

[Figure]

**Figure R2.** Evaluation of ozone in units mPa against WOUDC ozone sondes at Syowa station during August-October 2009. Black: ozone sonde, red: C-IFS-TS in the Revised model version, blue: C-IFS-TS in original GMDD version. Error bars denote the 1-sigma spread in the models and observations.

*p. 9, lines 31-32. If you made a difference plot between MIPAS and the simulations, then you might be able to say whether there is good agreement. As presented, the conclusion can't be drawn that there are 'small biases'. Near the tropical maximum the TS looks slightly better than the BASCOE-CTM. Again, assuming that some of the differences are due to dynamical fields or advection scheme, this comparison isn't very useful.*

As argued before, the inclusion of BASCOE-CTM is especially useful to diagnose if model biases arise due to different advection schemes or due to different chemistry schemes. To accommodate the reviewer's comments to better quantify the C-IFS versions, as compared to BASCOE-CTM we now include results from C-IFS-Atmos and C-IFS-T, and provide a more balanced discussion. Finally we have strengthened the evaluation of ozone with two new figures in the Supplementary Material: the quantitative comparison is strengthened by a new comparison of vertical profiles with Aura MLS (Fig. S7) and the discussion of Fig. 7 (top row) is confirmed with a corresponding evaluation also using Aura MLS (Fig. S8). The discussion of ozone on fig. 7 now reads:

"The evaluation of the zonal mean ozone **mixing ratios** against MIPAS observations shows good general agreement, Fig. 7, with **all four modelling experiments providing similar features. The tropical maximum of $O_3$ mixing ratio at 10 hPa is under-estimated in all experiments but to a larger extent in those which model stratospheric photochemistry explicitly (BASCOE CTM, C-IFS-TS, C-IFS-Atmos) than in C-IFS-T, in line with the evaluation against $O_3$ sondes for June-July-August (figure 4). The same evaluation against MLS observations provides exactly the same conclusions (figure S8, supplementary material)**."

*p.10, lines 5-9. What is the message here? The CIFS has a terrible high bias in nighttime NO2 and a large low bias in HNO3. Why is the CIFS simulation worse than BASCOE? There is no clear explanation here.*

We acknowledge that these results are not satisfying; unfortunately at current stage we do not have a clear explanation for this. Nevertheless, we want to highlight that the model performance has still improved compared to C-IFS-T, whose results we now include. Also we explicitly provide these figures to indicate current limitations of our model. We now write:

"**Even though a clear improvement compared to run C-IFS-T is found, further investigation is necessary to diagnose origins of the biases in night-time NO$_2$ above 10 hPa and in HNO$_3$ between 10 and 70 hPa.**"

Also in the conclusions section we include such a sentence.

*p. 10, lines 10-20 (Fig. 8). N2O and CH4 profiles do NOT assess vertical transport. Their profiles below _10 hPa represent a balance between the vertical and horizontal components of the residual mean circulation. That balance depends on latitude, that is, whether the profile is from the tropical upwelling region or somewhere in the midlatitudes (horizontal and vertical motions matter and so does mixing), or isolated inside the polar vortex (descent). Above 10 hPa, profiles are more strongly influenced by chemical loss so the 2 simulations should look very similar there. The CIFS-TS simulation tends to look worse than the BASCOE CTM or the observations between 10-50 hPa. This suggests circulation and/or mixing problems in the tropics and SH.*
*O3 at 20 hPa is strongly influence by chemistry, not just transport. These paragraphs indicate a lack of understanding of transport circulation and its diagnosis, as well as any understanding of what controls stratospheric ozone distributions.*

These two paragraphs are indeed rather vague and mistakenly use Figure 8 as a diagnostic for "vertical" transport. We thank the reviewer for pointing this out and setting us on the right track.
The reviewer's suggestion about circulation and/or mixing problems confirms that figure 8 is a preliminary yet valid diagnostic for transport processes in general, and that it indicates an unidentified issue for the representation of these processes in C-IFS. We do not think that circulation is the culprit because the BASCOE CTM is driven by meteorological fields which are the output of C-IFS. As noted above, the revised manuscript gives (at the beginning of section 3) a few more details about the modelling of transport in both models:

"**The BASCOE-CTM is driven by temperature, pressure and wind fields simulated by the C-IFS runs. However, while BASCOE adopts a flux-form advection scheme (Lin and Rood, 1996) the IFS uses the Semi-Lagrangian scheme for advection, explicitly accounts for horizontal diffusion and includes a parameterization for convection (ECMWF, 2015).**"

The revised manuscript also lists the relevant transport processes in the stratosphere along with a general reference on this topic (Shepherd, 2007) and states what specific pieces of C-IFS may be responsible for the problem(s). Since O$_3$ at 20 hPa is strongly influenced by both chemistry and transport, we stand with the statement that this transport issue "could also contribute" (directly) to the ozone biases noted below 10 hPa. But their attribution to an "excess of vertical transport" was clearly a mistake. No further statement can be made on this topic because further evaluation of stratospheric transport processes in C-IFS is beyond the scope of this paper. The two problematic paragraphs in section 4 have thus been re-written as follows:

[revised manuscript text omitted]